# On the Generalization Error of Stochastic Mirror Descent for Quadratically-Bounded Losses: an Improved Analysis

**Ta Duy Nguyen**
Department of Computer Science
Boston University
taduy@bu.edu

**Alina Ene**
Department of Computer Science
Boston University
aene@bu.edu

**Huy Le Nguyen**
Khoury College of Computer Sciences
Northeastern University
hu.nguyen@northeastern.edu

## Abstract

In this work, we revisit the generalization error of stochastic mirror descent for quadratically bounded losses studied in Telgarsky (2022). Quadratically bounded losses is a broad class of loss functions, capturing both Lipschitz and smooth functions, for both regression and classification problems. We study the high probability generalization for this class of losses on linear predictors in both realizable and non-realizable cases when the data are sampled IID or from a Markov chain. The prior work relies on an intricate coupling argument between the iterates of the original problem and those projected onto a bounded domain. This approach enables blackbox application of concentration inequalities, but also leads to suboptimal guarantees due in part to the use of a union bound across all iterations. In this work, we depart significantly from the prior work of Telgarsky (2022), and introduce a novel approach for establishing high probability generalization guarantees. In contrast to the prior work, our work directly analyzes the moment generating function of a novel supermartingale sequence and leverages the structure of stochastic mirror descent. As a result, we obtain improved bounds in all aforementioned settings. Specifically, in the realizable case and non-realizable case with light-tailed sub-Gaussian data, we improve the bounds by a $\log T$ factor, matching the correct rates of $1/T$ and $1/\sqrt{T}$, respectively. In the more challenging case of heavy-tailed polynomial data, we improve the existing bound by a $\text{poly } T$ factor.

## 1 Introduction

Along with convergence analysis of optimization methods, understanding the generalization of models trained by these methods on unseen data is an important question in machine learning. However, despite the number of works attempting to answer it, the problem has not been fully understood, even in the simplest setting of linear predictors constructed with the standard stochastic gradient/mirror descent. A great part of prior works [33, 15, 30, 31, 32] focus only on the generalization on linearly separable data and/or of models trained with specific losses with exponentially decaying tails such as logistic loss. The question of what we can guarantee beyond these settings remains open.

37th Conference on Neural Information Processing Systems (NeurIPS 2023).

Recently, [35] proposes a new approach to analyze the generalization error with *high probability* of stochastic mirror descent for a broad class of quadratically bounded losses, beyond the realizable setting. This class of losses encapsulates both Lipschitz and smooth functions, for both regression and classification problems. The obtained bounds complement existing in-expectation bounds [12] and nearly match the counterpart of convergence rates in optimization. While this result pushes forward the state of the art, the obtained guarantees do not completely resolve the problem. The central piece of the proposed approach is a "coupling" technique between the iterates of the original problem and those projected onto a bounded domain. In this technique, one first constrains the problem in a bounded domain with a well chosen diameter. The bounded domain diameter allows to apply concentration inequalities as a blackbox and obtain bounds in high probability. Then using an inductive argument and a union bound across all iterations, one can show that the iterates in the original problem coincide with the ones in the constrained problem. Due to the union bound, the success probability decreases from $1 - \delta$ to $1 - T\delta$, where $T$ is the number of iterations in the algorithm. This loss translates to a milder $\log T$ factor loss in the guarantee in the case of realizable data, and a more significant $\text{poly } T$ factor loss in the non-realizable setting when the data has polynomial tails. Thus a natural question arises of whether we can obtain a stronger analysis that closes these remaining gaps.

In this paper, we revisit these generalization bounds for quadratically bounded losses by [35]. We introduce a novel approach to analyze the generalization errors of stochastic mirror descent in both realizable and non-realizable cases when the data are sampled IID or from a Markov chain. In all these cases, we remove the need to use the union bound argument, thus preventing the loss in the success probability. This translates to the following improvements:

− In the realizable, and the non-realizable cases with sub-gaussian tailed data and Markovian data, we improve the bounds by a $\log T$ factor. This improvement comes from analyzing the moment generating function of a martingale difference sequence with well-chosen coefficients. In these cases, we also remove the necessity of using the coupling-based argument used in the same work by [35]. Instead, by solely making use of the problem structure, we arrive at the same conclusion that with high probability, the iterates of stochastic mirror descent for quadratically bounded losses behave as if the problem domain is bounded.

− In the non-realizable case with polynomial tailed data, we improve the existing bound by a $\text{poly } T$ factor. Due to the polynomial dependency on $\frac{1}{\delta}$, being able to maintain the same success probability through all iterations is crucial in this case. Unlike the previous work, we rely on a truncation technique. Using a more refined analysis of the truncated random variables, in combination with suitable concentration inequalities and the coupling technique, we improve the existing bounds significantly.

## 1.1 Related Work

Broadly speaking, there is a rich body of works in optimization and generalization that provide convergence guarantees and generalization bounds for stochastic methods. Earlier works often focus on in-expectation bounds [8, 24, 26, 18, 12, 3], and bounds in high probability [16, 28, 14, 13] for problems with bounded domains or under various additional assumptions such as strong convexity, noise with light tails. Recent developments for optimization [25, 10, 20, 23, 11, 17, 9, 19, 29, 22, 21] are able to handle unconstrained problems and relax these assumptions, but also require changes to the algorithm such as gradient clipping.

In generalization error analysis, specifically, a number of prior works, including [33, 15, 30, 31, 32], focus only on linearly separable data. Among these, [33, 15, 32] only deal with exponentially tailed losses while [30, 31] show generalization bounds for general smooth convex losses. Our work, similarly to [35], goes beyond the realizable setting and specific losses. We show high probability generalization bounds in both realizable and non-realizable settings for the broad class of quadratically bounded losses, for both regression and classification problems.

Other related works include the line of works that examine generalization errors via algorithmic stability. The works by [12, 5, 6, 2, 1] show the generalization error of an arbitrary algorithm via a quantity called uniform stability. By bounding this quantity for specific algorithms on a fixed training dataset, they derive generalization bounds. Our work focuses on a different setting where we assume the algorithm has access to a fresh data sample in each iteration. In this regard, the setting in our work,

as well as the prior work by [35], stays closer to the world of optimization. However, in contrast to the optimization world in which we commonly impose assumptions on the stochastic gradients (such as having bounded variance or sub-gaussian noise), we make assumptions on the data (such as sub-gaussian or polynomial tailed data). This difference introduces several challenges which we overcome in our work.

The main point of reference for this paper is the work by [35]. This work develops a "coupling" technique to bound the generalization error of stochastic mirror descent for quadratically bounded losses. This technique has been employed in prior works [10, 11, 9, 29, 27, 22] to obtain high probability convergence bounds of stochastic methods in optimization. Our work improves their results by using a different approach that takes a closer look at the mechanism of the concentration inequalities and leverages the problem structure. When the data are bounded or have sub-gaussian tails, analyzing the moment generating function of a novel martingale difference sequence allows us to maintain the same success probability, without using either the coupling technique or the union bound. This new analysis, however, does not change the observation by [35] that the iterates of the unconstrained and the constrained problems coincide with high probability. When the data have a polynomial tail, we rely on a truncation technique. In this case, the coupling technique is necessary but not the union bound, and we are still able to significantly improve the success rate.

In terms of techniques, the work by [21] for optimization is the closest to ours. In this work, the authors develop the whitebox approach to analyzing stochastic methods for optimization with light-tailed noise. In this work, we study generalization errors. Moreover, in all settings, our choice of martingale difference sequences and coefficients are a significant departure from the prior work. In particular, in [21] the choice of coefficients only depends on the problem parameters whereas in the realizable case, our coefficients depend also on the historical data. Our approach also allows for a flexible use of an induction argument without decreasing the success probability, while in [21] the bounds are simpler and can be easily achieved in a single step.

## 2 Preliminaries

In this section, we provide the general set up and necessary notations before analyzing stochastic mirror descent in the subsequent sections. Overall, we closely follow notations used in [35].

**Domain and norms**. In this work, we consider $\mathcal{X}$—the domain of the problem—to be a closed convex set or $\mathbb{R}^d$. We will use $\|\cdot\|$ to denote an arbitrary norm on $\mathcal{X}$ and let $\|\cdot\|_*$ be its dual norm. We define the Bregman divergence as $\mathbf{D}_\psi(w; v) = \psi(w) - \psi(v) - \langle \nabla \psi(v), w - v \rangle$ where $\psi : \mathbb{R}^d \to \mathbb{R}$ is a differentiable function that is 1-strongly convex with respect to the norm $\|\cdot\|$.

**Loss functions**. Each loss function $\ell : \mathbb{R} \times \mathbb{R} \to \mathbb{R}_{\geq 0}$ in our consideration can be written using a convex scalar function $\widetilde{\ell}$ in one of the two following forms: 1) $\ell(y, \widehat{y}) = \widetilde{\ell}(\text{sign}(y)\widehat{y})$ where $\text{sign}(y) = 1$ if $y \geq 0$ and $= -1$ otherwise; and 2) $\ell(y, \widehat{y}) = \widetilde{\ell}(y - \widehat{y})$. The first form captures classification losses and the second regression losses. We will assume that subgradients $\partial \ell$ of $\ell$ in the second argument always exist, and let $\ell'$ denote a subgradient in $\partial \ell$. For a function $f$, we also use $\|\partial f(w)\| := \sup \{\|g\| : g \in \partial f(w)\}$. We further make the following assumptions, introduced in [35] as quadratic boundedness and self-boundedness.

**Assumption 1**. We assume that $\ell$ is $(C_1, C_2)$-quadratically-bounded, for some constants $C_1, C_2 \geq 0$, i.e., for all $y, \widehat{y}$

$$|\ell'(y, \widehat{y})| \leq C_1 + C_2 (|y| + |\widehat{y}|).$$

This condition captures both classes of Lipschitz and smooth functions. Indeed, Lemma 1.2 from [35] shows that $\alpha$-Lipschitz functions are $(\alpha, 0)$-quadratically-bounded while $\beta$-smooth functions are $(\left|\partial \widetilde{\ell}(0)\right|, \beta)$-quadratically-bounded.

**Assumption 2**. In the realizable setting, we assume that $\ell$ is $\rho$-self-bounding, i.e., $\widetilde{\ell}$ satisfies $\widetilde{\ell}'(z)^2 \leq 2\rho \widetilde{\ell}(z)$ for all $z \in \mathbb{R}$.

The second assumption is a generalization of smoothness. This assumption is satisfied by smooth losses but also certain non-smooth losses such as the exponential loss. This condition is necessary in the current analysis to prove $1/T$ rates in the realizable setting. The readers can refer to [34, 35] for more detailed discussion on this assumption.

---

**Algorithm 1** Stochastic Mirror Descent

---

Input $w_0$, step size $\eta$
For $t$ in $1 \ldots T$
   $g_t \in \partial \ell_t(w_{t-1})$
   $w_t = \arg\min_{w \in \mathcal{X}} \{\langle \eta g_t, w \rangle + \mathbf{D}_\psi(w; w_{t-1})\}$

---

Assumptions 1 and 2 are satisfied by commonly used loss functions in machine learning. These include the logistic loss $\ell(y, \widehat{y}) = \ln(1 + \exp(-y\widehat{y}))$ and the squared loss $\ell(y, \widehat{y}) = \frac{1}{2}(y - \widehat{y})^2$ (see Lemma 1.4 in [35]).

For the loss function $\ell$ and the configuration $w$, and sample $(x, y)$ where $x$ denotes the attribute and $y$ the label, we will write $\ell_{x,y} = \ell(y, w^T x)$. We state the following crucial lemma which is the same as Lemma A.1 in [35], whose proof will be omitted.

**Lemma 1** (Lemma A.1 in ([35])). *Suppose $\ell$ is $(C_1, C_2)$-quadratically-bounded and $B_x \geq 0$ is given. Given $(x, y)$ such that $\max\{\|x\|_*, |y|\} \leq B_x$ and any $u, v$,*

$$\|\partial \ell_{x,y}(u)\|_* \leq B_x (C_1 + C_2 B_x (1 + \|u\|))$$
$$|\ell_{x,y}(u) - \ell_{x,y}(v)| \leq B_x \|u - v\| (C_1 + C_2 B_x (1 + \|u\|)).$$

**Risk, IID and Markovian data**. When sample $(x_i, y_i)$ arrives in iteration $i$ of an algorithm, we will use the notation $\ell_i(w) = \ell(y_i, w^T x_i)$. For an algorithm of $T$ iterations, we use $\mathcal{F}_t = \sigma((x_1, y_1), \ldots, (x_t, y_t))$ to denote the natural filtration up to and including time $t$. When the data are IID and generated from a distribution $\pi$, we define the risk

$$\mathcal{R}(w) = \mathbb{E}_{(x,y) \sim \pi}[\ell(y, w^T x)];$$

In contrast to IID data, Markovian data come from a stochastic process. This setting has also been considered in [4]. We let $P_s^t$ be the distribution of $(x_t, y_t)$ at iteration $t$ conditioned on $\mathcal{F}_s$. We make the following assumption regarding the uniform mixing time of the stochastic process. Note that similar assumptions have also appeared in [35, 4].

**Assumption 3**. We assume that for some $\epsilon, \tau \geq 0$ of our choice, there is a distribution $\pi$ such that

$$\sup_{t \in \mathbb{Z}_{\geq 0}} \sup_{\mathcal{F}_t} \mathrm{TV}\left(P_t^{t+\tau}, \pi\right) \leq \epsilon.$$

We refer to the triple $(\pi, \tau, \epsilon)$ as an approximate stationarity witness. We then define the risk according to the approximate stationary distribution $\pi$: $\mathcal{R}(w) = \mathbb{E}_{(x,y) \sim \pi}[\ell(y, w^T x)]$.

**Algorithm**. Stochastic Mirror Descent is given in Algorithm 1. In this algorithm, for the simplicity of the analysis, we consider a fixed step size $\eta$. In each iteration, we pick a subgradient $g_t \in \partial \ell_t(w_{t-1})$ and perform the update step.

We finally introduce a standard lemma used in the analysis of Stochastic Mirror Descent.

**Lemma 2.** *For $t \geq 0$ and $w_{\mathrm{ref}} \in \mathcal{X}$, we have*

$$\mathbf{D}_\psi(w_{\mathrm{ref}}; w_{t+1}) - \mathbf{D}_\psi(w_{\mathrm{ref}}; w_t) \leq \eta(\ell_{t+1}(w_{\mathrm{ref}}) - \ell_{t+1}(w_t)) + \frac{\eta^2}{2}\|g_{t+1}\|_*^2.$$

**Other notations**. We will use $w_{\mathrm{ref}}$ to refer to a comparator of interest. For the simplicity of the exposition, we let $D_0 = \mathbf{D}_\psi(w_{\mathrm{ref}}; w_0)$, and $\mathcal{R}^* = \inf_{v \in \mathcal{X}} \mathcal{R}(v)$. For a loss function $\ell$ that is $(C_1, C_2)$-quadratically-bounded, we let $C_4 = C_1 + C_2(1 + \|w_{\mathrm{ref}}\|)$.

## 3 Generalization bounds of SMD for IID data

In this section, we distinguish between two cases: the realizable case and the non-realizable case. In the realizable case, there exists an optimal solution $w^* \in \mathcal{X}$ such that $\mathcal{R}(w^*) = 0$. We will show that under mild assumptions, the risks of the solutions output by Algorithm 1 are bounded by $O(1/T)$. In the non-realizable case, we will show, on the other hand, a weaker statement that the excess risks of the solutions are bounded by $O(1/\sqrt{T})$.

### 3.1 Realizable case

In the realizable case, the comparator $w_{\text{ref}}$ is not necessarily the global minimizer. To show the $1/T$ rate, we will assume $w_{\text{ref}}$ satisfies $\mathcal{R}(w_{\text{ref}}) \leq \rho \mathbf{D}_\psi(w_{\text{ref}}; w_0)/T$ and that the loss at $w_{\text{ref}}$ is bounded. The guarantee for the iterates of Algorithm 1 is provided in Theorem 3.

**Theorem 3.** *Suppose $\ell$ is convex, $(C_1, C_2)$-quadratically-bounded, and $\rho$-self-bounding. Given $T$, $((x_t, y_t))_{t \leq T}$ are IID samples with $\max\{\|x_t\|_*, |y_t|\} \leq 1$ almost surely, $w_{\text{ref}}$ satisfies $\mathcal{R}(w_{\text{ref}}) \leq \rho \mathbf{D}_\psi(w_{\text{ref}}; w_0)/T$, and $\max_{t<T} \ell_{t+1}(w_{\text{ref}}) \leq C_3$ almost surely. Then for $\eta \leq \frac{1}{2\rho}$, with probability at least $1 - 2\delta$, for every $0 \leq k \leq T - 1$*

$$\frac{1}{k+1}\sum_{t=0}^{k} \mathcal{R}(w_t) + \frac{16\mathbf{D}_\psi(w_{\text{ref}}; w_{k+1})}{5(k+1)\eta} \leq \frac{C}{k+1} + 3\mathcal{R}(w_{\text{ref}}).$$

*where $C = \frac{16C_4}{5}\log\frac{1}{\delta}\sqrt{\frac{15}{4}D_0 + 4\eta\gamma C_3} + \left(\frac{6}{\eta}D_0 + \frac{32}{5}\gamma C_3\right)$ with $\gamma = \max\left\{1, \log\frac{1}{\delta}\right\}$.*

The analysis of Theorem 3 relies on the use of concentration inequalities. In contrast to existing works that utilize concentration inequalities as a blackbox, we will make use of the mechanism for proving concentration inequalities in order to obtain stronger guarantees. The type of concentration inequalities we consider are shown by analyzing the moment generating function of suitably chosen martingale sequences. We will use Lemma 14 (Appendix) which gives a basic inequality that bounds the moment generating function of a bounded random variable. To start the analysis, we use Lemma 2 and Assumption 2 to obtain

**Lemma 4.** *For all $t \geq 0$, we have*

$$\mathbf{D}_\psi(w_{\text{ref}}; w_{t+1}) - \mathbf{D}_\psi(w_{\text{ref}}; w_t) \leq \eta\ell_{t+1}(w_{\text{ref}}) - \frac{\eta}{2}\ell_{t+1}(w_t),$$

*and hence, $\mathbf{D}_\psi(w_{\text{ref}}; w_t) \leq \mathbf{D}_\psi(w_{\text{ref}}; w_0) + \eta\sum_{i=1}^{t}\ell_i(w_{\text{ref}}) = D_0 + \eta\sum_{i=1}^{t}\ell_i(w_{\text{ref}}).$*

First, let us pay attention to the term $\sum_{i=1}^{t}\ell_i(w_{\text{ref}})$. Recall that the terms $\ell_i(w_{\text{ref}})$ are non-negative and bounded by a constant $C_3$ almost surely. We can analyze the term $\sum_{i=1}^{T}\ell_i(w_{\text{ref}})$ which upper bounds all sums $\sum_{i=1}^{t}\ell_i(w_{\text{ref}})$ by studying its moment generating function (or via a concentration inequality). We state this bound in the next lemma and defer the proof to the appendix.

**Lemma 5.** *With probability at least $1 - \delta$, $\sum_{i=1}^{T}\ell_i(w_{\text{ref}}) \leq \frac{7}{4}T\mathcal{R}(w_{\text{ref}}) + C_3\log\frac{1}{\delta}$.*

Lemma 4 and lemma 5 and the assumption that $\mathcal{R}(w_{\text{ref}}) = O(1/T)$ imply that with probability at least $1 - \delta$, $\mathbf{D}_\psi(w_{\text{ref}}; w_t)$ is bounded. In other words, with probability at least $1 - \delta$, the iterates $w_t$ all lie in a bounded region. One could therefore proceed to assume that the problem domain is simply this bounded ball around $w_{\text{ref}}$. This is the basic idea behind the "coupling" technique demonstrated in [35]. However, the important question is how to obtain a bound for the risk of all iterates even when we are working with a problem with unbounded domain. Here, not paying close attention to the structure of the problem and the blackbox use of concentration inequalities lead to suboptimal bounds. On the other hand, as discussed above, a crucial novelty in our analysis is the choice of a supermartingale difference sequence, defined in the proof below. By working from first principles using moment generating function of this sequence, we derive two conclusions: 1) an improved risk bound can be obtained, and 2) the coupling technique is not necessary.

*Proof Sketch.* Towards bounding the risk $\sum_{t=0}^{k}\mathcal{R}(w_t)$, we define random variables

$$Z_t = \frac{1}{2}z_t\eta\left(\mathcal{R}(w_t) - \mathcal{R}(w_{\text{ref}}) - \ell_{t+1}(w_{\text{ref}})\right) + z_t\left(\mathbf{D}_\psi(w_{\text{ref}}; w_{t+1}) - \mathbf{D}_\psi(w_{\text{ref}}; w_t)\right)$$

$$- \frac{3}{16}z_t\eta\left(\mathcal{R}(w_{\text{ref}}) + \mathcal{R}(w_t)\right), \qquad \forall 0 \leq t \leq T - 1$$

$$\text{where } z_t = \frac{1}{\eta C_4\sqrt{2\eta\gamma C_3 + 2D_0 + 2\eta\sum_{i=1}^{t}\ell_i(w_{\text{ref}})}}; \quad \gamma = \max\left\{1, \log\frac{1}{\delta}\right\}$$

and we let $S_t = \sum_{i=0}^{t} Z_i;$ $\quad \forall 0 \leq t \leq T - 1$. The reason to define these variables is because from Lemma 4, we can bound

$$\mathbb{E}\left[\exp\left(Z_t\right) \mid \mathcal{F}_t\right] \times \exp\left(\frac{3}{16} z_t \eta \left(\mathcal{R}(w_{\text{ref}}) + \mathcal{R}(w_t)\right)\right)$$

$$\leq \mathbb{E}\left[\exp\left(\frac{1}{2} z_t \eta \left(\mathcal{R}(w_t) - \mathcal{R}(w_{\text{ref}}) - \ell_{t+1}\left(w_{\text{ref}}\right)\right) + z_t \left(\eta \ell_{t+1}\left(w_{\text{ref}}\right) - \frac{\eta}{2}\ell_{t+1}(w_t)\right)\right) \mid \mathcal{F}_t\right]$$

$$= \mathbb{E}\left[\exp\left(\frac{1}{2} z_t \eta \left(\mathcal{R}(w_t) - \mathcal{R}(w_{\text{ref}}) + \ell_{t+1}(w_{\text{ref}}) - \ell_{t+1}(w_t)\right)\right) \mid \mathcal{F}_t\right]$$

where now inside the expectation, we have the term $\mathcal{R}(w_t) - \mathcal{R}(w_{\text{ref}}) + \ell_{t+1}(w_{\text{ref}}) - \ell_{t+1}(w_t)$ which has expectation 0. Let $C_4 = C_1 + C_2(1 + \|w_{\text{ref}}\|)$, we can bound

$$\left|\frac{\eta}{2}\left(\mathcal{R}(w_t) - \mathcal{R}(w_{\text{ref}}) + \ell_{t+1}(w_{\text{ref}}) - \ell_{t+1}(w_t)\right)\right| \leq \eta C_4 \|w_{\text{ref}} - w_t\|$$

and $z_t \leq \frac{1}{\eta C_4 \|w_{\text{ref}} - w_t\|}$. Now we can apply Lemma 14 (Appendix) to bound

$$\mathbb{E}\left[\exp\left(Z_t\right) \mid \mathcal{F}_t\right] \times \exp\left(\frac{3}{16} z_t \eta \left(\mathcal{R}(w_{\text{ref}}) + \mathcal{R}(w_t)\right)\right)$$

$$\leq \exp\left(\frac{3}{4}\frac{1}{4} z_t^2 \eta^2 \mathbb{E}\left[\left(\mathcal{R}(w_t) - \mathcal{R}(w_{\text{ref}}) + \ell_{t+1}(w_{\text{ref}}) - \ell_{t+1}(w_t)\right)^2 \mid \mathcal{F}_t\right]\right)$$

$$\leq \exp\left(\frac{3}{16} z_t^2 \eta^2 \mathbb{E}\left[\left(\ell_{t+1}(w_{\text{ref}}) - \ell_{t+1}(w_t)\right)^2 \mid \mathcal{F}_t\right]\right)$$

$$\leq \exp\left(\frac{3}{16} z_t^2 \eta^2 C_4 \|w_{\text{ref}} - w_t\| \, \mathbb{E}\left[\ell_{t+1}(w_{\text{ref}}) + \ell_{t+1}(w_t) \mid \mathcal{F}_t\right]\right)$$

$$\leq \exp\left(\frac{3}{16} z_t \eta \left(\mathcal{R}(w_{\text{ref}}) + \mathcal{R}(w_t)\right)\right)$$

Therefore $\mathbb{E}\left[\exp\left(Z_t\right) \mid \mathcal{F}_t\right] \leq 1$ and hence $\left(\exp\left(S_t\right)\right)_{t \geq 0}$ is a supermartingale. By Ville's inequality, we have with probability at least $1 - \delta$, for all $0 \leq k \leq T - 1$

$$\sum_{t=0}^{k} Z_t \leq \log \frac{1}{\delta}$$

Expanding this inequality, in combination with Lemma 5, we obtain the conclusion. $\qquad \square$

*Remark* 6. The new analysis does not change the conclusion observed in [35]—that is, with high probability, the iterate sequence $(w_t)_{t \geq 0}$ behaves as if the domain of the problem is bounded. We improve the probability that this event happens.

## 3.2 Non-realizable case

In the non-realizable case, we do not aim for $1/T$ but only $1/\sqrt{T}$ rates. Hence we do not assume that the comparator $w_{\text{ref}}$ satisfies $\mathcal{R}(w_{\text{ref}}) \leq \rho \mathbf{D}_\psi(w_{\text{ref}}; w_0)/T$ but rather the following assumption on the excess risk:

**Assumption 4.** Let $\mathcal{R}^* = \inf_{v \in \mathcal{X}} \mathcal{R}(v)$, assume that $\mathcal{R}(w_{\text{ref}}) - \mathcal{R}^* \leq \frac{\mathbf{D}_\psi(w_{\text{ref}}; w_0)}{\sqrt{T}}$.

We also relax the assumption on the data samples. In the previous case, the data are bounded, i.e $\{\|x\|_*, |y|\} \leq 1$ a.s. We will consider in this section two settings, one when the data come from a sub-Gaussian distribution and one when the data distribution has a polynomial tail.

**Remark on Theorem 10 in [35]**. There seems to be an issue with the proof of Theorem 10 in [35]. The proof uses a variant of Azuma's inequality ([37], Problem 3.11) which allows the ranges of the random variables to not be specified up front. However, when bounding the range of relevant random variables, the proof uses $Z_{t+1}$, which is correlated with the data $\|x_{t+1}\|_*$ and $y_{t+1}$. Thus the condition of Azuma's inequality is not satisfied. We do not see an immediate way to resolve this problem. In the following, we consider separately the two cases of sub-Gaussian and polynomial

tailed data for which we use different proof techniques to show the error bounds. In the first case of IID data with sub-Gaussian tails, we proceed by bounding the moment generating function of a well-chosen martingale sequence. In the second case of IID data with polynomial tails, we introduce a truncation technique. In both cases, we are able to obtained better bounds compared with [35].

### 3.2.1 IID data with sub-Gaussian tails

We will show the following guarantee:

**Theorem 7.** *Suppose $\ell$ is convex, $(C_1, C_2)$-quadratically-bounded. Given $T$, $((x_t, y_t))_{t \leq T}$ are IID samples with $Q_t = \max\left\{1, \|x_t\|_*^2, |y_t|^2\right\}$ and there exists $\sigma \geq 0$ such that for all $\lambda$*

$$\max\left\{\mathbb{E}\left[\exp\left(\lambda\left(Q_t^2 - \mathbb{E}\left[Q_t^2\right]\right)\right)\right], \mathbb{E}\left[\exp\left(\lambda\left(Q_t - \mathbb{E}\left[Q_t\right]\right)\right)\right]\right\} \leq \exp\left(\lambda^2\sigma^2\right)$$

*Let $\mu_1 = \mathbb{E}[Q_t]$ and $\mu_2 = \mathbb{E}\left[Q_t^2\right]$. Suppose that $w_{\mathrm{ref}}$ satisfies Assumption 4. Then for $\eta \leq \frac{1}{4C_2\sqrt{T\mu_2 + 2\sigma\sqrt{T\log\frac{1}{\delta}}}}$, with probability at least $1 - 2\delta$, for every $0 \leq k \leq T - 1$*

$$\frac{1}{k+1}\sum_{t=0}^{k}\left(\mathcal{R}(w_t) - \mathcal{R}^*\right) + \frac{\mathbf{D}_\psi\left(w_{\mathrm{ref}}; w_{k+1}\right)}{\eta(k+1)} \leq \frac{R^2}{\eta(k+1)}$$

*where $R^2 = 16C_4^2\left(\sigma^2 + 4\mu_1^2\right)\log\frac{1}{\delta}\eta^2 T + 4D_0(1 + \eta\sqrt{T}) + 4\eta^2 C_4^2\left(T\mu_2 + 2\sigma\sqrt{T\log\frac{1}{\delta}}\right) = O(1)$.*

*Remark 8.* For zero-mean sub-Gaussian variable $X$, the definition $\mathbb{E}\left[\exp\left(\lambda X\right)\right] \leq \exp\left(\lambda^2\sigma^2\right)$ for all $\lambda$ is equivalent to $\mathbb{E}\left[\exp\left(\lambda^2 X^2\right)\right] \leq \exp\left(\lambda^2\sigma^2\right)$ for all $0 \leq \lambda \leq \frac{1}{\sigma}$ (see [38]). The lemma below shows a property of sub-Gaussian variables under scaling and translating. First let us consider $\sum_{t=1}^{T} Q_t^2$. Similar to Lemma 5, by bounding the moment generating function of this term, we have the following (see also Section B4 in [35]).

**Lemma 9.** *With probability at least $1 - \delta$, $\sum_{t=1}^{T} Q_t^2 \leq T\mu_2 + 2\sigma\sqrt{T\log\frac{1}{\delta}}$.*

*Proof of Theorem 7.* The proof of this Theorem uses the technique developed in [21]. We will also analyze the moment generating function of a suitable martingale sequence. However, the choice of the coefficients will differ significantly from the previous proof. In this case the structure of the problem is deeply integrated into the analysis of the martingale. We define

$$Z_t = z_t\eta\left(\mathcal{R}(w_t) - \mathcal{R}(w_{\mathrm{ref}})\right) + z_t\left(\mathbf{D}_\psi\left(w_{\mathrm{ref}}; w_{t+1}\right) - \mathbf{D}_\psi\left(w_{\mathrm{ref}}; w_t\right)\right)$$
$$- \frac{1}{2}z_t\eta^2\|g_{t+1}\|_*^2 - 4z_t^2\eta^2 C_4^2\left(\sigma^2 + 4\mu_1^2\right)\mathbf{D}_\psi\left(w_{\mathrm{ref}}; w_t\right) \qquad \forall 0 \leq t \leq T - 1$$

where $z_t = \dfrac{1}{4\eta^2 C_4^2\left(\sigma^2 + 4\mu_1^2\right)(T + t + 1)}$ $\qquad \forall -1 \leq t \leq T - 1$

and let $S_t = \sum_{i=0}^{t} Z_i$; $\quad \forall 0 \leq t \leq T - 1$. By Lemma 2, we have

$$Z_t + 4z_t^2\eta^2 C_4^2\left(\sigma^2 + 4\mu_1^2\right)\mathbf{D}_\psi\left(w_{\mathrm{ref}}; w_t\right) \leq z_t\eta\left(\mathcal{R}(w_t) - \mathcal{R}(w_{\mathrm{ref}}) + \ell_{t+1}\left(w_{\mathrm{ref}}\right) - \ell_{t+1}\left(w_t\right)\right)$$

where we have $\mathbb{E}\left[\left(\mathcal{R}(w_t) - \mathcal{R}(w_{\mathrm{ref}}) + \ell_{t+1}\left(w_{\mathrm{ref}}\right) - \ell_{t+1}\left(w_t\right)\right)\right] = 0$, and using the same notation $C_4 = C_1 + C_2(1 + \|w_{\mathrm{ref}}\|)$, by Lemma 1,

$$\left|\left(\mathcal{R}(w_t) - \mathcal{R}(w_{\mathrm{ref}}) + \ell_{t+1}\left(w_{\mathrm{ref}}\right) - \ell_{t+1}\left(w_t\right)\right)\right|$$
$$\leq \left|\ell_{t+1}\left(w_{\mathrm{ref}}\right) - \ell_{t+1}\left(w_t\right)\right| + \left|\mathcal{R}(w_t) - \mathcal{R}(w_{\mathrm{ref}})\right|$$
$$\leq \left|\ell_{t+1}\left(w_{\mathrm{ref}}\right) - \ell_{t+1}\left(w_t\right)\right| + \mathbb{E}\left[\left|\ell_{x,y}(w_t) - \ell_{x,y}(w_{\mathrm{ref}})\right|\right]$$
$$\leq \left(Q_t + \mu_1\right)\|w_{\mathrm{ref}} - w_t\|C_4 = \left(\left(Q_t - \mu_1\right) + 2\mu_1\right)\|w_{\mathrm{ref}} - w_t\|C_4$$

Hence applying Lemma 15, we have

$$\mathbb{E}\left[\exp\left(Z_t\right) \mid \mathcal{F}_t\right]\exp\left(4z_t^2\eta^2 C_4^2\left(\sigma^2 + 4\mu_1^2\right)\mathbf{D}_\psi\left(w_{\mathrm{ref}}; w_t\right)\right)$$
$$= \mathbb{E}\left[\exp\left(z_t\eta_t\left(\mathcal{R}(w_t) - \mathcal{R}(w_{\mathrm{ref}}) + \ell_{t+1}\left(w_{\mathrm{ref}}\right) - \ell_{t+1}\left(w_t\right)\right)\right) \mid \mathcal{F}_t\right]$$

$$\leq \exp\left(2z_t^2\eta^2C_4^2\left\|w_{\mathrm{ref}}-w_t\right\|^2\left(\sigma^2+4\mu_1^2\right)\right)$$

$$\leq \exp\left(4z_t^2\eta^2C_4^2\left(\sigma^2+4\mu_1^2\right)\mathbf{D}_\psi\left(w_{\mathrm{ref}};w_t\right)\right)$$

Therefore $\mathbb{E}\left[\exp\left(Z_t\right)\mid\mathcal{F}_t\right]\leq 1$ and hence $\left(\exp\left(S_t\right)\right)_{t\geq 0}$ is a supermartingale. By Ville's inequality, we have with probability at least $1-\delta$, for all $0\leq k\leq T-1$

$$\sum_{t=0}^{k}Z_t\leq\log\frac{1}{\delta}$$

Expanding this inequality we have

$$\sum_{t=0}^{k}z_t\eta\mathcal{R}(w_t)+z_k\mathbf{D}_\psi\left(w_{\mathrm{ref}};w_{k+1}\right)$$

$$\leq\log\frac{1}{\delta}+z_{-1}D_0+\eta\mathcal{R}(w_{\mathrm{ref}})\sum_{t=0}^{k}z_t+\frac{1}{2}\sum_{t=0}^{k}z_t\eta^2\left\|g_{t+1}\right\|_*^2$$

$$+\sum_{t=0}^{k}\underbrace{\left(z_t+4z_t^2\eta^2C_4^2\left(\sigma^2+4\mu_1^2\right)-z_{t-1}\right)}_{\leq 0}\mathbf{D}_\psi\left(w_{\mathrm{ref}};w_t\right)$$

$$\overset{(a)}{\leq}\log\frac{1}{\delta}+z_{-1}D_0+\eta\mathcal{R}(w_{\mathrm{ref}})\sum_{t=0}^{k}z_t+\frac{1}{2}\sum_{t=0}^{k}z_t\eta^2\left\|g_{t+1}\right\|_*^2$$

where for $(a)$, by the choice of $z_t = \frac{1}{4\eta^2C_4^2\left(\sigma^2+4\mu_1^2\right)(T+1+t)}$ we have $z_{t-1}-z_t\geq 4z_t^2\eta^2C_4^2\left(\sigma^2+4\mu_1^2\right)$. We highlight that this is where the structure of the problem comes into play. That is, by setting appropriate coefficients, we can leverage gain in the distance in the martingale difference sequence $\left((z_t-z_{t-1})\mathbf{D}_\psi\left(w_{\mathrm{ref}};w_t\right)\right)$ to cancel out the loss from bounding the moment generating function $\left(4z_t^2\eta^2C_4^2\left(\sigma^2+4\mu_1^2\right)\mathbf{D}_\psi\left(w_{\mathrm{ref}};w_t\right)\right)$. Another important property of the sequence $(z_t)$ is that it is a decreasing sequence and $\frac{z_t}{z_k}\leq 2$ for all $t,k$. Hence we have

$$\eta\sum_{t=0}^{k}\left(\mathcal{R}(w_t)-\mathcal{R}^*\right)+\mathbf{D}_\psi\left(w_{\mathrm{ref}};w_{k+1}\right)$$

$$\leq 4C_4^2\left(\sigma^2+4\mu_1^2\right)\log\frac{1}{\delta}\eta^2\left(T+1+k\right)+2D_0+2\left(\mathcal{R}(w_{\mathrm{ref}})-\mathcal{R}^*\right)\eta(k+1)+\eta^2\sum_{t=0}^{k}\left\|g_{t+1}\right\|_*^2.$$

Combined with Lemma 9, with probability at least $1-2\delta$, for all $0\leq k\leq T-1$

$$\eta\sum_{t=0}^{k}\left(\mathcal{R}(w_t)-\mathcal{R}^*\right)+\mathbf{D}_\psi\left(w_{\mathrm{ref}};w_{k+1}\right)$$

$$\leq 4C_4^2\left(\sigma^2+4\mu_1^2\right)\log\frac{1}{\delta}\eta^2\left(T+1+k\right)+2D_0+2\left(\mathcal{R}(w_{\mathrm{ref}})-\mathcal{R}^*\right)\eta(k+1)+\eta^2\sum_{t=0}^{k}\left\|g_{t+1}\right\|_*^2;$$

and $\sum_{t=1}^{k+1}Q_t^2\leq T\mu_2+2\sigma\sqrt{T\log\frac{1}{\delta}}$

Conditioned on this event, we will prove by induction that

$$\mathbf{D}_\psi\left(w_{\mathrm{ref}};w_k\right)$$

$$\leq R^2:=16C_4^2\left(\sigma^2+4\mu_1^2\right)\log\frac{1}{\delta}\eta^2T+4D_0+4D_0\eta\sqrt{T}+4\eta^2C_4^2\left(T\mu_2+2\sigma\sqrt{T\log\frac{1}{\delta}}\right)$$

For the base case $k=0$, it is trivial. Suppose for all $t\leq k$ we have $\mathbf{D}_\psi\left(w_{\mathrm{ref}};w_t\right)\leq R^2$, now we prove for $t=k+1$. By Lemma 1,

$$\eta^2\sum_{t=0}^{k}\left\|g_{t+1}\right\|_*^2\leq\eta^2\sum_{t=0}^{k}Q_{t+1}^2\left(C_1+C_2\left(1+\left\|w_t\right\|\right)\right)^2\leq\eta^2\sum_{t=0}^{k}Q_{t+1}^2\left(C_4+C_2\left\|w_t-w_{\mathrm{ref}}\right\|\right)^2$$

$$\leq 2\eta^2 C_4^2 \sum_{t=1}^{k+1} Q_t^2 + 2\eta^2 C_2^2 \sum_{t=0}^{k} Q_{t+1}^2 \|w_t - w_{\mathrm{ref}}\|^2$$

$$\leq \eta^2 \left(2C_4^2 + 4C_2^2 R^2\right)\left(T\mu_2 + 2\sigma\sqrt{T\log\frac{1}{\delta}}\right)$$

Therefore

$$\mathbf{D}_\psi\left(w_{\mathrm{ref}}; w_{k+1}\right) \leq 8C_4^2\left(\sigma^2 + 4\mu_1^2\right)\log\frac{1}{\delta}\eta^2 T + 2D_0 + 2\left(\mathcal{R}(w_{\mathrm{ref}}) - \mathcal{R}^*\right)\eta(k+1)$$

$$+ \eta^2\left(2C_4^2 + 4C_2^2 R^2\right)\left(T\mu_2 + 2\sigma\sqrt{T\log\frac{1}{\delta}}\right)$$

$$\leq \frac{R^2}{2} + 4\eta^2 C_2^2\left(T\mu_2 + 2\sigma\sqrt{T\log\frac{1}{\delta}}\right)R^2 \leq R^2.$$

Finally we obtain, $\eta\sum_{t=0}^{k}\left(\mathcal{R}(w_t) - \mathcal{R}^*\right) + \mathbf{D}_\psi\left(w_{\mathrm{ref}}; w_{k+1}\right) \leq R^2$, as needed. $\qquad\square$

### 3.2.2 IID data with polynomial tails

**Theorem 10.** *Suppose $\ell$ is convex, $(C_1, C_2)$-quadratically bounded. Given $T$, $((x_t, y_t))_{t\leq T}$ are IID samples with $Q_t = \max\left\{1, \|x_t\|_*^2, |y_t|^2\right\}$ and for some $p \geq 2$ there exists $M \geq \frac{p}{e}$ such that for all $\lambda$*

$$\max\left\{\sup_{2\leq r\leq 2p}\left\{\mathbb{E}\left[|Q_t - \mathbb{E}\left[Q_t\right]|^r\right]\right\}, \sup_{2\leq r\leq p}\left\{\mathbb{E}\left[\left|Q_t^2 - \mathbb{E}\left[Q_t^2\right]\right|^r\right]\right\}\right\} \leq M$$

*Let $\mu_1 = \mathbb{E}\left[Q_t\right]$ and $\mu_2 = \mathbb{E}\left[Q_t^2\right]$. Suppose that $w_{\mathrm{ref}}$ satisfies Assumption 4. Then for $\eta \leq \dfrac{1}{C_2\sqrt{6\left(T\mu_2 + 2M\sqrt{T}\left(\frac{2}{\delta}\right)^{\frac{1}{p}}\right)}}$, with probability at least $1 - 3\delta$, for every $0 \leq k \leq T - 1$*

$$\frac{1}{k+1}\sum_{t=0}^{k}\left(\mathcal{R}(w_t) - \mathcal{R}^*\right) + \frac{\mathbf{D}_\psi\left(w_{\mathrm{ref}}; w_{k+1}\right)}{\eta(k+1)} \leq \frac{R^2}{2\eta(k+1)}$$

*where $R = \max\left\{\sqrt{6\left(D_0\left(1 + \eta\sqrt{T}\right) + \eta^2 C_4^2\left(T\mu_2 + 2M\sqrt{T}\left(\frac{2}{\delta}\right)^{\frac{1}{p}}\right)\right)}, 6\left(\frac{2}{3}\gamma\left(7\left(\frac{MT}{\delta}\right)^{1/2p} + 2\mu_1\right) + \sqrt{\log\frac{2}{\delta}T\mu_2\eta C_4}\right)\right\} = O(1)$, $\gamma = \max\left\{1, \log\frac{2}{\delta}\right\}$.*

*Remark* 11. Since $p \geq 2$, the rate is $O\left(\frac{1}{T^{1/2}}\log\frac{1}{\delta} + \frac{1}{T^{3/4}}\left(\frac{1}{\delta}\right)^{\frac{1}{2p}}\right)$. This rate improves over the $O\left(\left(\frac{1}{T^{1/2}} + \frac{1}{T^{3/4}}\left(\frac{T}{\delta}\right)^{\frac{1}{2p}}\right)\log\frac{T}{\delta}\right)$ rate by [35] by a polynomial factor $T^{\frac{1}{2p}}\log\frac{T}{\delta}$ in the high probability regime where $\delta = \frac{1}{\mathrm{poly}(T)}$.

We will give a proof sketch for this theorem. The full proof is deferred to the appendix.

*Proof Sketch.* The heavy tailed distribution of the data does not allow us to analyze the moment generating function. In this case, we rely on the coupling technique as in [35]. Since it is not possible to apply Azuma's inequality due to the bounds on the variables being not measurable, and the variables are heavy tailed, we use truncation technique. We define,

$$v_t = \arg\min_{\|w - w_{\mathrm{ref}}\|\leq R}\left\{\langle\eta_t g_t(v_{t-1}), w\rangle + \mathbf{D}_\psi\left(w; v_{t-1}\right)\right\}$$

where we use $g_t(v_{t-1})$ to denote the gradient at $v_{t-1}$ using the same data point $(x_t, y_t)$ when computing $w_t$ and we define

$$U_t = \left(\mathcal{R}(v_t) - \mathcal{R}(w_{\mathrm{ref}}) + \ell_{t+1}\left(w_{\mathrm{ref}}\right) - \ell_{t+1}\left(v_t\right)\right)$$

$$P_t = \begin{cases} U_t & \text{if } |U_t| \leq (A + 2\mu_1) RC_4 \\ (A + 2\mu_1) RC_4 \text{sign}(U_t) & \text{otherwise} \end{cases}$$

$$\text{where } A = \left(\frac{MT}{\delta}\right)^{1/2p} \text{ and } B_t = U_t - P_t.$$

We can write

$$\sum_{t=0}^{k} U_t = \sum_{t=0}^{k} (P_t - \mathbb{E}[P_t \mid \mathcal{F}_t]) + \sum_{t=0}^{k} \mathbb{E}[P_t \mid \mathcal{F}_t] + \sum_{t=0}^{k} B_t$$

We bound $\sum_{t=0}^{k} (P_t - \mathbb{E}[P_t \mid \mathcal{F}_t])$ by applying Freedman's inequality. The terms $\sum_{t=0}^{k} \mathbb{E}[P_t \mid \mathcal{F}_t]$ and $\sum_{t=0}^{k} B_t$ are both the bias terms can be bounded by analyzing the tail of the distribution and Markov's inequality. We also use Lemma 12 to bound $\sum_{t=0}^{k} \|g_{t+1}(v_t)\|_*^2$. Finally, using the induction technique, we can prove that $w_t = v_t$ with high probability and obtain the desired result. $\qquad \square$

**Lemma 12** (Lemma A.5 from [35]). *With probability* $\geq 1 - \delta$, $\sum_{t=1}^{T} Q_t^2 \leq T\mu_2 + 2M\sqrt{T} \left(\frac{2}{\delta}\right)^{\frac{1}{p}}$.

## 4 Generalization bounds of SMD for Markovian data

The final result we present in this paper is the following theorem for the case when the data are sampled from a Markov chain.

**Theorem 13.** *Suppose* $\ell$ *is convex,* $(C_1, C_2)$-*quadratically bounded. Given* $T$, $((x_t, y_t))_{t \leq T}$ *are sampled from a Markov chain with* $\max\left\{\|x_t\|_*^2, |y_t|^2\right\} \leq 1$ *and* $\left(\pi, \tau, \epsilon = \frac{1}{\sqrt{T}}\right)$ *is an approximate stationarity witness. Suppose that* $w_{\text{ref}}$ *satisfies Assumption 4. Then for* $\eta \leq \frac{1}{2C_2\sqrt{T(1+2\tau)}}$, *with probability at least* $1 - \tau\delta$, *for every* $0 \leq k \leq T - 1$

$$\frac{1}{k+1} \sum_{t=0}^{k} (\mathcal{R}(w_t) - \mathcal{R}^*) + \frac{\mathbf{D}_\psi(w_{\text{ref}}; w_{k+1})}{\eta(k+1)} \leq \frac{R^2}{2\eta(k+1)}.$$

*where* $R = \max\left\{\sqrt{6\left(2D_0 + 2\eta D_0 \sqrt{T} + 16\eta^2 C_4^2 T\tau \log \frac{1}{\delta} + 2T\eta^2 C_4^2 + 4\eta^2 \tau T C_4^2\right)}, 6(2\eta\tau C_4 + 2\eta C_4 \epsilon T + 4\eta\tau C_4)\right\} = O(1)$ *and* $C_4 = C_1 + C_2(1 + \|w_{\text{ref}}\|)$.

We will give a proof sketch for this theorem.

*Proof Sketch.* The proof of this Theorem follow similarly to that of Theorem 7. The difference here is we need to bound $\tau$ different martingale difference sequences in the form of

$$\mathbb{E}\left[\ell_{\tau(i+1)+j}(w_{\text{ref}}) \mid \mathcal{F}_{\tau i+j}\right] - \mathbb{E}\left[\ell_{\tau(i+1)+j}(w_{\tau i+j}) \mid \mathcal{F}_{\tau i+j}\right] + \ell_{\tau(i+1)+j}(w_{\text{ref}}) - \ell_{\tau(i+1)+j}(w_{\tau i+j})$$

for $0 \leq j \leq \tau - 1, 0 \leq i \leq \frac{T-1-j}{\tau}$. We also need the assumption on the approximate stationarity witness to see that

$$|\mathcal{R}(w_t) - \mathcal{R}(w_{\text{ref}}) - \mathbb{E}[\ell_{t+\tau}(w_{\text{ref}}) \mid \mathcal{F}_t] + \mathbb{E}[\ell_{t+\tau}(w_t) \mid \mathcal{F}_t]| \leq C_4 R\epsilon.$$

Now we only need the union bound over $\tau$ sequences, instead of all iterations. The success probability will decrease from $1 - \delta$ to $1 - \tau\delta$. $\qquad \square$

## 5 Conclusion

In this paper, we show a new approach to analyze the generalization error of SMD for quadratically bounded losses. Our approach improves a logarithmic factor for the realizable setting and non-realizable setting with light tailed data and a poly $T$ factor for the non-realizable setting with polynomial tailed data from the prior work by [35]. An inherent limitation of the current approach is the assumption that we can obtain a fresh sample in each iteration, whereas the setting with finite training data is still not well understood. In the realizable setting, we require that the data is bounded, as opposed to more relaxed assumptions in the non-realizable settings. We leave the question of resolving these issues for future works.

## Acknowledgement

TDN and AE were supported in part by NSF CAREER grant CCF-1750333, NSF grant III-1908510, and an Alfred P. Sloan Research Fellowship. HN was supported by NSF CAREER grant CCF-1750716 and NSF grant 2311649.

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

# A  Concentration Inequalities

**Lemma 14.** *Let $X$ be a random variable such that $\mathbb{E}[X] = 0$ and $|X| \leq R$ almost surely. Then for $0 \leq \lambda \leq \frac{1}{R}$*

$$\mathbb{E}\left[\exp\left(\lambda X\right)\right] \leq \exp\left(\frac{3}{4}\lambda^2\mathbb{E}\left[X^2\right]\right).$$

The following lemma is similar to Lemma 2.2 in [21].

**Lemma 15.** *Suppose that $Q$ satisfies for all $0 \leq \lambda \leq \frac{1}{\sigma}$, $\mathbb{E}\left[\exp\left(\lambda^2 Q^2\right)\right] \leq \exp\left(\lambda^2\sigma^2\right)$. Then for variable $X$ such that $\mathbb{E}[X] = 0$ and $|X| \leq a\left(Q + b\right)$ for some $a \geq 0$ then for all $\lambda \geq 0$*

$$\mathbb{E}\left[\exp\left(\lambda X\right)\right] \leq \exp\left(2\lambda^2 a^2\left(\sigma^2 + b^2\right)\right).$$

*In particular, if $b = 0$ we can have a tighter constant:* $\mathbb{E}\left[\exp\left(\lambda X\right)\right] \leq \exp\left(\lambda^2 a^2\sigma^2\right)$.

*Proof.* We consider $\mathbb{E}\left[\exp\left(\lambda X\right)\right]$

If $0 \leq \lambda \leq \frac{1}{\sqrt{2}a\sigma}$ then using $\exp\left(x\right) \leq x + \exp\left(x^2\right)$

$$\begin{aligned}
\mathbb{E}\left[\exp\left(\lambda X\right)\right] &\leq \mathbb{E}\left[\exp\left(\lambda^2 X^2\right)\right] \\
&\leq \mathbb{E}\left[\exp\left(\lambda^2 a^2\left(Q + b\right)^2\right)\right] \\
&\leq \mathbb{E}\left[\exp\left(2\lambda^2 a^2 Q^2 + 2\lambda^2 a^2 b^2\right)\right] \\
&\leq \exp\left(2\lambda^2 a^2 b^2\right)\mathbb{E}\left[\exp\left(2\lambda^2 a^2 Q^2\right)\right] \\
&\leq \exp\left(2\lambda^2 a^2\left(\sigma^2 + b^2\right)\right)
\end{aligned}$$

Otherwise $\frac{1}{\sigma} \leq \lambda\sqrt{2}a$

$$\begin{aligned}
\mathbb{E}\left[\exp\left(\lambda X\right)\right] &\leq \mathbb{E}\left[\exp\left(\lambda^2 a^2\sigma^2 + \frac{X^2}{4a^2\sigma^2}\right)\right] \\
&\leq \exp\left(\lambda^2 a^2\sigma^2\right)\mathbb{E}\left[\exp\left(\frac{\left(Q + b\right)^2}{4\sigma^2}\right)\right] \\
&\leq \exp\left(\lambda^2 a^2\sigma^2\right)\mathbb{E}\left[\exp\left(\frac{Q^2 + b^2}{2\sigma^2}\right)\right] \\
&\leq \exp\left(\lambda^2 a^2\sigma^2\right)\exp\left(\frac{b^2}{2\sigma^2}\right)\exp\left(\frac{1}{2}\right) \\
&\leq \exp\left(\lambda^2 a^2\sigma^2\right)\exp\left(\lambda^2 a^2 b^2\right)\exp\left(\lambda^2 a^2\sigma^2\right) \\
&\leq \exp\left(2\lambda^2 a^2\left(\sigma^2 + b^2\right)\right).
\end{aligned}$$

$\square$

**Theorem 16** (Freedman's inequality [7, 36]). *Let $(X_t)_{t \geq 1}$ be a martingale difference sequence. Assume that there exists a constant $c$ such that $|X_t| \leq c$ almost surely for all $t \geq 1$ and define $\sigma_t^2 = \mathbb{E}\left[X_t^2 \mid X_{t-1}, \ldots, X_1\right]$. Then for all $b > 0$, $F > 0$ and $T \geq 1$*

$$\Pr\left[\exists T \geq 1 : \left|\sum_{t=1}^T X_t\right| > b \text{ and } \sum_{t=1}^T \sigma_t^2 \leq F\right] \leq 2\exp\left(-\frac{b^2}{2F + 2cb/3}\right).$$

# B  Missing Proofs

*Proof of Lemma 2.* Using the optimality condition

$$\langle \eta g_{t+1} + \nabla\psi(w_{t+1}) - \nabla\psi(w_t), w_{\text{ref}} - w_{t+1}\rangle \geq 0$$

we have

$$\langle \eta g_{t+1}, w_t - w_{\mathrm{ref}} \rangle = \langle \eta g_{t+1}, w_{t+1} - w_{\mathrm{ref}} \rangle + \langle \eta g_{t+1}, w_t - w_{t+1} \rangle$$
$$\leq \langle \nabla \psi(w_{t+1}) - \nabla \psi(w_t), w_{\mathrm{ref}} - w_{t+1} \rangle + \langle \eta g_{t+1}, w_t - w_{t+1} \rangle$$
$$= \mathbf{D}_\psi \left( w_{\mathrm{ref}}; w_t \right) - \mathbf{D}_\psi \left( w_{\mathrm{ref}}; w_{t+1} \right) - \mathbf{D}_\psi \left( w_{t+1}; w_t \right) + \langle \eta g_{t+1}, w_t - w_{t+1} \rangle$$
$$\leq \mathbf{D}_\psi \left( w_{\mathrm{ref}}; w_t \right) - \mathbf{D}_\psi \left( w_{\mathrm{ref}}; w_{t+1} \right) - \frac{1}{2} \| w_t - w_{t+1} \|^2 + \langle \eta g_{t+1}, w_t - w_{t+1} \rangle$$
$$\leq \mathbf{D}_\psi \left( w_{\mathrm{ref}}; w_t \right) - \mathbf{D}_\psi \left( w_{\mathrm{ref}}; w_{t+1} \right) + \frac{\eta^2}{2} \| g_{t+1} \|_*^2$$

Hence

$$\mathbf{D}_\psi \left( w_{\mathrm{ref}}; w_{t+1} \right) - \mathbf{D}_\psi \left( w_{\mathrm{ref}}; w_t \right) \leq \langle \eta g_{t+1}, w_{\mathrm{ref}} - w_t \rangle + \frac{\eta^2}{2} \| g_{t+1} \|_*^2$$
$$\leq \eta \left( \ell_{t+1} \left( w_{\mathrm{ref}} \right) - \ell_{t+1} \left( w_t \right) \right) + \frac{\eta^2}{2} \| g_{t+1} \|_*^2$$

as needed. $\square$

*Proof of Lemma 4.* We have

$$\mathbf{D}_\psi \left( w_{\mathrm{ref}}; w_{t+1} \right) - \mathbf{D}_\psi \left( w_{\mathrm{ref}}; w_t \right) \leq \eta \left( \ell_{t+1} \left( w_{\mathrm{ref}} \right) - \ell_{t+1} \left( w_t \right) \right) + \frac{\eta^2}{2} \| g_{t+1} \|_*^2$$
$$\leq \eta \left( \ell_{t+1} \left( w_{\mathrm{ref}} \right) - \ell_{t+1} \left( w_t \right) \right) + \frac{\eta^2}{2} \ell'_{t+1}(w_t)^2$$
$$\leq \eta \left( \ell_{t+1} \left( w_{\mathrm{ref}} \right) - \ell_{t+1} \left( w_t \right) \right) + \eta^2 \rho \ell_{t+1}(w_t)$$
$$= \eta \ell_{t+1} \left( w_{\mathrm{ref}} \right) - \frac{\eta}{2} \ell_{t+1}(w_t) \leq \eta \ell_{t+1} \left( w_{\mathrm{ref}} \right).$$

Summing up, we have, for any $0 \leq t \leq T$

$$\mathbf{D}_\psi \left( w_{\mathrm{ref}}; w_t \right) \leq \mathbf{D}_\psi \left( w_{\mathrm{ref}}; w_0 \right) + \eta \sum_{i=1}^t \ell_i \left( w_{\mathrm{ref}} \right) = D_0 + \eta \sum_{i=1}^t \ell_i \left( w_{\mathrm{ref}} \right).$$

$\square$

*Proof of Lemma 5.* We have $\left| \ell_i \left( w_{\mathrm{ref}} \right) - \mathcal{R}(w_{\mathrm{ref}}) \right| \leq \max \left\{ \ell_i \left( w_{\mathrm{ref}} \right), \mathcal{R}(w_{\mathrm{ref}}) \right\} \leq C_3$ thus by lemma 14, for $\lambda \leq \frac{1}{C_3}$

$$\mathbb{E} \left[ \exp \left( \lambda \left( \ell_i \left( w_{\mathrm{ref}} \right) - \mathcal{R}(w_{\mathrm{ref}}) \right) \right) \right]$$
$$\leq \exp \left( \frac{3}{4} \lambda^2 \mathbb{E} \left[ \left( \ell_i \left( w_{\mathrm{ref}} \right) - \mathcal{R}(w_{\mathrm{ref}}) \right)^2 \right] \right)$$
$$\overset{(a)}{\leq} \exp \left( \frac{3}{4} \lambda^2 \mathbb{E} \left[ \ell_i \left( w_{\mathrm{ref}} \right)^2 \right] \right)$$
$$\overset{(b)}{\leq} \exp \left( \frac{3}{4} \lambda^2 C_3 \mathcal{R}(w_{\mathrm{ref}}) \right) \leq \exp \left( \frac{3}{4} \lambda \mathcal{R}(w_{\mathrm{ref}}) \right),$$

where for $(a)$ we use $\mathbb{E} \left[ (X - \mathbb{E}[X])^2 \right] \leq \mathbb{E}[X^2]$ and for $(b)$ we use $\ell_i \left( w_{\mathrm{ref}} \right) \leq C_3$. Since $\ell_i \left( w_{\mathrm{ref}} \right)$ are independent random variables, we have

$$\mathbb{E} \left[ \exp \left( \lambda \sum_{i=1}^T \left( \ell_i \left( w_{\mathrm{ref}} \right) - \mathcal{R}(w_{\mathrm{ref}}) \right) \right) \right] = \mathbb{E} \left[ \prod_{i=1}^T \exp \left( \lambda \left( \ell_i \left( w_{\mathrm{ref}} \right) - \mathcal{R}(w_{\mathrm{ref}}) \right) \right) \right]$$
$$= \prod_{i=1}^T \mathbb{E} \left[ \exp \left( \lambda \left( \ell_i \left( w_{\mathrm{ref}} \right) - \mathcal{R}(w_{\mathrm{ref}}) \right) \right) \right] \leq \prod_{i=1}^T \exp \left( \frac{3}{4} \lambda \mathcal{R}(w_{\mathrm{ref}}) \right) = \exp \left( \frac{3}{4} \lambda T \mathcal{R}(w_{\mathrm{ref}}) \right).$$

Hence by Markov's inequality

$$\Pr\left[\lambda\sum_{i=1}^{T}\left(\ell_i\left(w_{\mathrm{ref}}\right)-\mathcal{R}(w_{\mathrm{ref}})\right)\geq\frac{3}{4}\lambda T\mathcal{R}(w_{\mathrm{ref}})+\log\frac{1}{\delta}\right]$$

$$=\Pr\left[\exp\left(\lambda\sum_{i=1}^{T}\left(\ell_i\left(w_{\mathrm{ref}}\right)-\mathcal{R}(w_{\mathrm{ref}})\right)\right)\geq\frac{1}{\delta}\exp\left(\frac{3}{4}\lambda T\mathcal{R}(w_{\mathrm{ref}})\right)\right]$$

$$\leq\frac{\mathbb{E}\left[\exp\left(\lambda\sum_{i=1}^{T}\left(\ell_i\left(w_{\mathrm{ref}}\right)-\mathcal{R}(w_{\mathrm{ref}})\right)\right)\right]}{\frac{1}{\delta}\exp\left(\frac{3}{4}\lambda T\mathcal{R}(w_{\mathrm{ref}})\right)}\leq\delta$$

Choose $\lambda=\frac{1}{C_3}$ we have with probability at least $1-\delta$

$$\sum_{i=1}^{T}\left(\ell_i\left(w_{\mathrm{ref}}\right)-\mathcal{R}(w_{\mathrm{ref}})\right)\leq\frac{3}{4}T\mathcal{R}(w_{\mathrm{ref}})+C_3\log\frac{1}{\delta}.$$

$\square$

*Proof of Theorem 3.* Towards bounding the risk $\sum_{t=0}^{k}\mathcal{R}(w_t)$, we define random variables

$$Z_t=\frac{1}{2}z_t\eta\left(\mathcal{R}(w_t)-\mathcal{R}(w_{\mathrm{ref}})-\ell_{t+1}\left(w_{\mathrm{ref}}\right)\right)+z_t\left(\mathbf{D}_\psi\left(w_{\mathrm{ref}};w_{t+1}\right)-\mathbf{D}_\psi\left(w_{\mathrm{ref}};w_t\right)\right)$$

$$-\frac{3}{16}z_t\eta\left(\mathcal{R}(w_{\mathrm{ref}})+\mathcal{R}(w_t)\right);\qquad\forall 0\leq t\leq T-1$$

where $z_t=\dfrac{1}{\eta C_4\sqrt{2\eta\gamma C_3+2D_0+2\eta\sum_{i=1}^{t}\ell_i\left(w_{\mathrm{ref}}\right)}};\quad\gamma=\max\left\{1,\log\frac{1}{\delta}\right\}$

and $S_t=\displaystyle\sum_{i=0}^{t}Z_i;\qquad\forall 0\leq t\leq T-1$

The reason to define these variables is because from Lemma 4, we can bound

$$\mathbb{E}\left[\exp\left(Z_t\right)\mid\mathcal{F}_t\right]\times\exp\left(\frac{3}{16}z_t\eta\left(\mathcal{R}(w_{\mathrm{ref}})+\mathcal{R}(w_t)\right)\right)$$

$$\leq\mathbb{E}\left[\exp\left(\frac{1}{2}z_t\eta\left(\mathcal{R}(w_t)-\mathcal{R}(w_{\mathrm{ref}})-\ell_{t+1}\left(w_{\mathrm{ref}}\right)\right)+z_t\left(\eta\ell_{t+1}\left(w_{\mathrm{ref}}\right)-\frac{\eta}{2}\ell_{t+1}(w_t)\right)\right)\mid\mathcal{F}_t\right]$$

$$=\mathbb{E}\left[\exp\left(\frac{1}{2}z_t\eta\left(\mathcal{R}(w_t)-\mathcal{R}(w_{\mathrm{ref}})+\ell_{t+1}(w_{\mathrm{ref}})-\ell_{t+1}(w_t)\right)\right)\mid\mathcal{F}_t\right]$$

where now inside the expectation, we have the term $\mathcal{R}(w_t)-\mathcal{R}(w_{\mathrm{ref}})+\ell_{t+1}(w_{\mathrm{ref}})-\ell_{t+1}(w_t)$ which has expectation $0$. This reminds us of Lemma 14. To use this lemma, we notice that, by the assumption that the samples are IID with $\max\left\{\|x\|_*,|y|\right\}\leq 1$ and Lemma 1,

$$|\ell_{x,y}(w_{\mathrm{ref}})-\ell_{x,y}(w_t)|\leq\|w_{\mathrm{ref}}-w_t\|\underbrace{\left(C_1+C_2(1+\|w_{\mathrm{ref}}\|)\right)}_{C_4}$$

We also have

$$|\mathcal{R}(w_t)-\mathcal{R}(w_{\mathrm{ref}})|=|\mathbb{E}\left[\ell_{x,y}(w_{\mathrm{ref}})-\ell_{x,y}(w_t)\right]|\leq C_4\|w_{\mathrm{ref}}-w_t\|$$

Therefore

$$\left|\frac{\eta}{2}\left(\mathcal{R}(w_t)-\mathcal{R}(w_{\mathrm{ref}})+\ell_{t+1}(w_{\mathrm{ref}})-\ell_{t+1}(w_t)\right)\right|\leq\eta C_4\|w_{\mathrm{ref}}-w_t\|$$

By the choice of $z_t$ we have

$$z_t\leq\frac{1}{\eta C_4\sqrt{2\eta C_3+2D_0+2\eta\sum_{i=1}^{t}\ell_i\left(w_{\mathrm{ref}}\right)}}\leq\frac{1}{\eta C_4\sqrt{2\mathbf{D}_\psi\left(w_{\mathrm{ref}};w_t\right)}}\leq\frac{1}{\eta C_4\|w_{\mathrm{ref}}-w_t\|}$$

Now we can apply Lemma 14 to bound

$$\mathbb{E}\left[\exp\left(Z_t\right) \mid \mathcal{F}_t\right] \times \exp\left(\frac{3}{16} z_t \eta \left(\mathcal{R}(w_{\mathrm{ref}}) + \mathcal{R}(w_t)\right)\right)$$

$$\leq \exp\left(\frac{3}{4}\frac{1}{4} z_t^2 \eta^2 \mathbb{E}\left[\left(\mathcal{R}(w_t) - \mathcal{R}(w_{\mathrm{ref}}) + \ell_{t+1}(w_{\mathrm{ref}}) - \ell_{t+1}(w_t)\right)^2 \mid \mathcal{F}_t\right]\right)$$

$$\leq \exp\left(\frac{3}{16} z_t^2 \eta^2 \mathbb{E}\left[\left(\ell_{t+1}(w_{\mathrm{ref}}) - \ell_{t+1}(w_t)\right)^2 \mid \mathcal{F}_t\right]\right)$$

$$\leq \exp\left(\frac{3}{16} z_t^2 \eta^2 C_4 \left\|w_{\mathrm{ref}} - w_t\right\| \mathbb{E}\left[\ell_{t+1}(w_{\mathrm{ref}}) + \ell_{t+1}(w_t) \mid \mathcal{F}_t\right]\right)$$

$$\leq \exp\left(\frac{3}{16} z_t \eta \left(\mathcal{R}(w_{\mathrm{ref}}) + \mathcal{R}(w_t)\right)\right)$$

Therefore $\mathbb{E}\left[\exp\left(Z_t\right) \mid \mathcal{F}_t\right] \leq 1$ and hence $\left(\exp\left(S_t\right)\right)_{t \geq 0}$ is a supermartingale. By Ville's inequality, we have with probability at least $1 - \delta$, for all $0 \leq k \leq T - 1$

$$\sum_{t=0}^{k} Z_t \leq \log\frac{1}{\delta}$$

Expanding this inequality, we obtain

$$\sum_{t=0}^{k} \frac{5}{16} z_t \eta \mathcal{R}(w_t) + z_k \mathbf{D}_\psi\left(w_{\mathrm{ref}}; w_{k+1}\right)$$

$$\leq \log\frac{1}{\delta} + z_0 \mathbf{D}_\psi\left(w_{\mathrm{ref}}; w_0\right) + \frac{11}{16}\eta\mathcal{R}(w_{\mathrm{ref}})\sum_{t=0}^{k} z_t + \frac{1}{2}\sum_{t=0}^{k} z_t \eta \ell_{t+1}(w_{\mathrm{ref}})$$

$$+ \sum_{t=1}^{k} \underbrace{(z_t - z_{t-1})}_{\leq 0} \mathbf{D}_\psi\left(w_{\mathrm{ref}}; w_t\right) \tag{1}$$

$$\overset{(a)}{\leq} \log\frac{1}{\delta} + z_0 D_0 + \frac{11}{16}\eta\mathcal{R}(w_{\mathrm{ref}})(k+1)z_0 + \frac{1}{2}\sum_{t=0}^{k} \frac{\eta\ell_{t+1}(w_{\mathrm{ref}})}{\eta C_4 \sqrt{2\eta C_3 + 2D_0 + 2\eta\sum_{i=1}^{t}\ell_i\left(w_{\mathrm{ref}}\right)}}$$

$$\overset{(b)}{\leq} \log\frac{1}{\delta} + z_0 D_0 + \frac{11}{16}\eta\mathcal{R}(w_{\mathrm{ref}})(k+1)z_0 + \frac{1}{2\eta C_4}\sum_{t=0}^{k} \frac{\eta\ell_{t+1}(w_{\mathrm{ref}})}{\sqrt{2D_0 + 2\eta\sum_{i=1}^{t+1}\ell_i\left(w_{\mathrm{ref}}\right)}} \tag{2}$$

For $(a)$ we use the fact that $(z_t)$ is a decreasing sequence and $\mathcal{R}(w_{\mathrm{ref}}) \leq \frac{\rho D_0}{T}$. For $(b)$ we use the assumption $\ell_{t+1}\left(w_{\mathrm{ref}}\right) \leq C_3$. Now notice that we can write $\frac{\eta\ell_{t+1}(w_{\mathrm{ref}})}{\sqrt{2D_0 + 2\eta\sum_{i=1}^{t+1}\ell_i(w_{\mathrm{ref}})}} \leq$ $\sqrt{2D_0 + 2\eta\sum_{i=1}^{t+1}\ell_i\left(w_{\mathrm{ref}}\right)} - \sqrt{2D_0 + 2\eta\sum_{i=1}^{t}\ell_i\left(w_{\mathrm{ref}}\right)}$ and sum over $t$ we obtain

$$\frac{5}{16} z_k \eta \sum_{t=0}^{k} \mathcal{R}(w_t) + z_k \mathbf{D}_\psi\left(w_{\mathrm{ref}}; w_{k+1}\right)$$

$$\leq \log\frac{1}{\delta} + \frac{11(k+1)\mathcal{R}(w_{\mathrm{ref}})}{16C_4\sqrt{2\eta\gamma C_3 + 2D_0}} + \frac{1}{\sqrt{2\eta}C_4}\sqrt{D_0 + \eta\sum_{i=1}^{k+1}\ell_i\left(w_{\mathrm{ref}}\right)}$$

Hence

$$\sum_{t=0}^{k} \mathcal{R}(w_t) + \frac{16}{5\eta}\mathbf{D}_\psi\left(w_{\mathrm{ref}}; w_{k+1}\right)$$

$$\leq \frac{16C_4}{5}\left(\log\frac{1}{\delta} + \frac{11(k+1)\mathcal{R}(w_{\mathrm{ref}})}{16C_4\sqrt{2\eta\gamma C_3 + 2D_0}} + \frac{1}{\sqrt{2\eta}C_4}\sqrt{D_0 + \eta\sum_{i=1}^{T}\ell_i\left(w_{\mathrm{ref}}\right)}\right)$$

$$\times \sqrt{2\eta\gamma C_3 + 2D_0 + 2\eta \sum_{i=1}^{T} \ell_i\left(w_{\mathrm{ref}}\right)}$$

By Lemma 5, with probability at least $1 - \delta$ we have

$$\sum_{i=1}^{T} \ell_i\left(w_{\mathrm{ref}}\right) \leq \frac{7}{4} T \mathcal{R}(w_{\mathrm{ref}}) + C_3 \log \frac{1}{\delta} \leq \frac{7}{4}\rho D_0 + C_3 \gamma$$

Therefore with probability at least $1 - 2\delta$

$$\sum_{t=0}^{k} \mathcal{R}(w_t) + \frac{16}{5\eta} \mathbf{D}_\psi\left(w_{\mathrm{ref}}; w_{k+1}\right)$$

$$\leq \left( \frac{16 C_4}{5} \log \frac{1}{\delta} + \frac{11(k+1)}{5\sqrt{2\eta\gamma C_3 + 2D_0}} \mathcal{R}(w_{\mathrm{ref}}) + \frac{8}{5\eta}\sqrt{\frac{15}{4} D_0 + 2\eta\gamma C_3} \right) \sqrt{\frac{15}{4} D_0 + 4\eta\gamma C_3}$$

$$\leq \frac{16 C_4}{5} \log \frac{1}{\delta} \sqrt{\frac{15}{4} D_0 + 4\eta\gamma C_3} + \left( \frac{6}{\eta} D_0 + \frac{32}{5}\gamma C_3 \right) + 3(k+1)\mathcal{R}(w_{\mathrm{ref}}).$$

which gives us the conclusion. $\qquad\square$

*Proof of Theorem 10.* First we consider the bounded domain case. Let

$$v_t = \arg \min_{\|w - w_{\mathrm{ref}}\| \leq R} \left\{ \langle \eta_t g_t(v_{t-1}), w \rangle + \mathbf{D}_\psi\left(w; v_{t-1}\right) \right\}$$

where we use $g_t(v_{t-1})$ to denote the gradient at $v_{t-1}$ using the same data point $(x_t, y_t)$ when computing $w_t$ and we choose

$$R = \max\left\{ \sqrt{6\left( D_0 + \eta^2 C_4^2 \left( T\mu_2 + 2M\sqrt{T}\left(\frac{2}{\delta}\right)^{\frac{1}{p}} \right) \right)}, \right.$$

$$\left. 6\left( \frac{2}{3}\gamma \left( 7\left(\frac{MT}{\delta}\right)^{1/2p} + 2\mu_1 \right) + \sqrt{\log \frac{2}{\delta} T\mu_2} \right) \eta C_4 \right\}$$

We have

$$|(\mathcal{R}(v_t) - \mathcal{R}(w_{\mathrm{ref}}) + \ell_{t+1}\left(w_{\mathrm{ref}}\right) - \ell_{t+1}\left(v_t\right))|$$
$$\leq |\ell_{t+1}\left(w_{\mathrm{ref}}\right) - \ell_{t+1}\left(v_t\right)| + |\mathcal{R}(v_t) - \mathcal{R}(w_{\mathrm{ref}})|$$
$$\leq (Q_t + \mu_1)\|w_{\mathrm{ref}} - v_t\| C_4 \leq (Q_t + \mu_1) R C_4 \qquad (3)$$

Let us define the following variables

$$U_t = (\mathcal{R}(v_t) - \mathcal{R}(w_{\mathrm{ref}}) + \ell_{t+1}\left(w_{\mathrm{ref}}\right) - \ell_{t+1}\left(v_t\right))$$

$$P_t = \begin{cases} U_t & \text{if } |U_t| \leq (A + 2\mu_1) R C_4 \\ (A + 2\mu_1) R C_4 \mathrm{sign}\left(U_t\right) & \text{otherwise} \end{cases}$$

where $A = \left(\frac{MT}{\delta}\right)^{1/2p}$ and $B_t = U_t - P_t$.

In words, $U_t$ is the variable of our interest and $P_t$ is the truncated version of $U_t$ and $B_t$ is the bias. We would want to control these terms in order to bound $\sum_{t=0}^{k} U_t$. We start with the following decomposition

$$\sum_{t=0}^{k} U_t = \sum_{t=0}^{k} (P_t - \mathbb{E}\left[P_t \mid \mathcal{F}_t\right]) + \sum_{t=0}^{k} \mathbb{E}\left[P_t \mid \mathcal{F}_t\right] + \sum_{t=0}^{k} B_t$$

First, we consider the term $\sum_{t=0}^{k} \mathbb{E}\left[P_t \mid \mathcal{F}_t\right]$.

$$\mathbb{E}\left[P_t \mid \mathcal{F}_t\right] = \mathbb{E}\left[P_t - U_t \mid \mathcal{F}_t\right] \le \mathbb{E}\left[|P_t - U_t| \mid \mathcal{F}_t\right]$$

$$=\mathbb{E}\Bigg[ |P_t - U_t| \left(\mathbf{1}\left[|U_t| \le (A + 2\mu_1)RC_4\right]\right.$$

$$\left. + \sum_{k=2}^{\infty} \mathbf{1}\left[(k-1)ARC_4 + 2\mu_1 RC_4 \le |U_t| \le kARC_4 + 2\mu_1 RC_4\right]\right)\Bigg]$$

$$=\mathbb{E}\left[\sum_{k=2}^{\infty} |P_t - U_t|\,\mathbf{1}\left[(k-1)ARC_4 + 2\mu_1 RC_4 \le |U_t| \le kARC_4 + 2\mu_1 RC_4\right]\right]$$

$$\le \sum_{k=2}^{\infty} \left(kARC_4 + 2\mu_1 RC_4 - (A + 2\mu_1)RC_4\right) RC_4 \mathbb{E}\left[\mathbf{1}\left[|U_t| \ge (k-1)ARC_4 + 2\mu_1 RC_4\right]\right]$$

$$\le \sum_{k=1}^{\infty} kARC_4 \mathbb{E}\left[\mathbf{1}\left[(Q_t + \mu_1)RC_4 \ge kARC_4 + 2\mu_1 RC_4\right]\right] \qquad \text{(due to 3)}$$

$$= \sum_{k=1}^{\infty} kARC_4 \Pr\left[Q_t \ge kA + \mu_1\right] \le ARC_4 \sum_{k=1}^{\infty} k \Pr\left[|Q_t - \mu_1|^{2p} \ge (kA)^{2p}\right]$$

$$\le ARC_4 \sum_{k=1}^{\infty} \frac{Mk}{k^{2p}A^{2p}} = A^{1-2p}RC_4 M \sum_{k=1}^{\infty} k^{1-2p} \le 2A^{1-2p}RC_4 M$$

where the last inequality is because $p \ge 2$. We obtain

$$\sum_{t=0}^{k} \mathbb{E}\left[P_t \mid \mathcal{F}_t\right] \le 2A^{1-2p}RC_4 MT$$

The term $\sum_{t=0}^{k} B_t \le \sum_{t=0}^{k} |B_t| \le \sum_{t=0}^{T-1} |B_t|$ will be bounded by Markov inequality. From the above deduction,

$$\mathbb{E}\left[\sum_{t=0}^{T-1} |B_t|\right] = \sum_{t=0}^{T-1} \mathbb{E}\left[|B_t|\right] = \sum_{t=0}^{T-1} \mathbb{E}\left[\mathbb{E}\left[|U_t - P_t| \mid \mathcal{F}_t\right]\right] \le 2A^{1-2p}RC_4 MT$$

With probability at least $1 - \delta$,

$$\sum_{t=0}^{T-1} |B_t| \le 2TA^{1-2p}RC_4 M \frac{1}{\delta} = 2RC_4 A^{1-2p}\left(\frac{MT}{\delta}\right)$$

Finally, we will use Freedman's inequality to bound the remaining term $\sum_{t=0}^{k} \left(P_t - \mathbb{E}\left[P_t \mid \mathcal{F}_t\right]\right)$. First, notice that

$$\mathbb{E}\left[|P_t - \mathbb{E}\left[P_t \mid \mathcal{F}_t\right]|^2 \mid \mathcal{F}_t\right] \le \mathbb{E}\left[P_t^2 \mid \mathcal{F}_t\right]$$

$$\le \mathbb{E}\left[U_t^2 \mid \mathcal{F}_t\right] \le \mathbb{E}\left[\left(\ell_{t+1}(w_{\mathrm{ref}}) - \ell_{t+1}(v_t)\right)^2 \mid \mathcal{F}_t\right]$$

$$\le R^2 C_4^2 \mathbb{E}\left[Q_t^2\right] \le R^2 C_4^2 \mu_2.$$

We have $\left(P_t - \mathbb{E}\left[P_t \mid \mathcal{F}_t\right]\right)$ is a martingale difference sequence with $|P_t - \mathbb{E}\left[P_t \mid \mathcal{F}_t\right]| \le 2(A + 2\mu_1)RC_4$. We can apply Freedman's inequality,

$$\Pr\left[\exists k \ge 0 : \left|\sum_{t=0}^{k} P_t - \mathbb{E}\left[P_t \mid \mathcal{F}_t\right]\right| > a \text{ and } \sum_{t=0}^{k} \mathbb{E}\left[|P_t - \mathbb{E}\left[P_t \mid \mathcal{F}_t\right]|^2 \mid \mathcal{F}_t\right] \le F\right]$$

$$\le 2\exp\left(\frac{-2a^2}{2F + 4(A + 2\mu_1)RC_4 a/3}\right)$$

If we select

$$F = T\mu_2 R^2 C_4^2$$

$$\text{and } a = \frac{2}{3} \log \frac{2}{\delta} \left( A + 2\mu_1 \right) RC_4 + RC_4 \sqrt{\log \frac{2}{\delta} T \mu_2}$$

we obtain with probability at least $1 - \delta$, for all $k \geq 0$

$$\sum_{t=0}^{k} P_t - \mathbb{E}\left[ P_t \mid \mathcal{F}_t \right] \leq \frac{2}{3} \log \frac{2}{\delta} \left( A + 2\mu_1 \right) RC_4 + RC_4 \sqrt{\log \frac{2}{\delta} T \mu_2}$$

Therefore with probability at least $1 - 3\delta$ we have the following event $E$ : for all $k \geq 0$

$$\sum_{t=0}^{k} U_t \leq \frac{2}{3} \log \frac{2}{\delta} \left( A + 2\mu_1 \right) RC_4 + RC_4 \sqrt{\log \frac{2}{\delta} T \mu_2} + 4RC_4 A^{1-2p} \left( \frac{MT}{\delta} \right)$$

$$\leq \frac{2}{3} \gamma \left( 7 \left( \frac{MT}{\delta} \right)^{1/2p} + 2\mu_1 \right) RC_4 + RC_4 \sqrt{\log \frac{2}{\delta} T \mu_2}$$

$$\text{and } \sum_{t=1}^{k+1} Q_t^2 \leq T\mu_2 + 2M\sqrt{T} \left( \frac{2}{\delta} \right)^{\frac{1}{p}}.$$

where we denote $\gamma = \max\left\{ 1, \log \frac{2}{\delta} \right\}$. Furthermore

$$\frac{\eta^2}{2} \sum_{t=0}^{k} \| g_{t+1}(v_t) \|_*^2 \leq \frac{\eta^2}{2} \sum_{t=0}^{k} Q_{t+1}^2 \left( C_1 + C_2 \left( 1 + \| v_t \| \right) \right)^2$$

$$\leq \frac{\eta^2}{2} \sum_{t=0}^{k} Q_{t+1}^2 \left( C_4 + C_2 \| v_t - w_{\text{ref}} \| \right)^2$$

$$\leq \eta^2 C_4^2 \sum_{t=1}^{k+1} Q_t^2 + \eta^2 C_2^2 \sum_{t=0}^{k} Q_{t+1}^2 \| v_t - w_{\text{ref}} \|^2$$

$$\leq \eta^2 \left( C_4^2 + C_2^2 R^2 \right) \left( T\mu_2 + 2M\sqrt{T} \left( \frac{2}{\delta} \right)^{\frac{1}{p}} \right)$$

Now we will proceed by induction to show that conditioned on the event $E$, $w_t = v_t$. For the base case, we have $w_0 = v_0$. Suppose that we have $w_t = v_t$ for all $t \leq k$. We will show that $w_{k+1} = v_{k+1}$. From Lemma 2, we have

$$\sum_{t=0}^{k} \eta \left( \mathcal{R}(w_t) - \mathcal{R}^* \right) + \mathbf{D}_\psi \left( w_{\text{ref}}; w_{k+1} \right)$$

$$\leq D_0 + \sum_{t=0}^{k} \eta \left( \mathcal{R}(w_{\text{ref}}) - \mathcal{R}^* \right)$$

$$+ \eta \sum_{t=0}^{k} \left( \mathcal{R}(w_t) - \mathcal{R}(w_{\text{ref}}) + \ell_{t+1} \left( w_{\text{ref}} \right) - \ell_{t+1} \left( w_t \right) \right) + \frac{\eta^2}{2} \sum_{t=0}^{k} \| g_{t+1} \|_*^2$$

$$\leq D_0 + \eta \sqrt{T} D_0 + \eta \sum_{t=0}^{k} U_t + \frac{\eta^2}{2} \sum_{t=0}^{k} \| g_{t+1}(v_t) \|_*^2$$

$$\leq D_0 \left( 1 + \eta \sqrt{T} \right)$$

$$+ \left( \frac{2}{3} \gamma \left( 7 \left( \frac{MT}{\delta} \right)^{1/2p} + 2\mu_1 \right) + \sqrt{\log \frac{2}{\delta} T \mu_2} \right) \eta RC_4$$

$$+ \eta^2 \left( C_4^2 + C_2^2 R^2 \right) \left( T\mu_2 + 2M\sqrt{T} \left( \frac{2}{\delta} \right)^{\frac{1}{p}} \right)$$

$$\leq \frac{R^2}{2}$$

Thus $\|w_{k+1} - w_{\text{ref}}\| \leq R$. And thus $w_{k+1} = v_{k+1}$. Finally, we can conclude that with probability at least $1 - 3\delta$, for all $0 \leq k \leq T - 1$

$$\frac{1}{k+1} \sum_{t=0}^{k} \left( \mathcal{R}(w_t) - \mathcal{R}(w_{\text{ref}}) \right) + \frac{\mathbf{D}_\psi \left( w_{\text{ref}}; w_{k+1} \right)}{\eta \left( k+1 \right)} \leq \frac{R^2}{2\eta \left( k+1 \right)}.$$

$\square$

*Proof of Theorem 13.* For $0 \leq j \leq \tau - 1$, we define

$$Z_i^j = z_{\tau i+j} \eta \left( \mathbb{E} \left[ \ell_{\tau(i+1)+j} \left( w_{\text{ref}} \right) \mid \mathcal{F}_{\tau i+j} \right] - \mathbb{E} \left[ \ell_{\tau(i+1)+j} \left( w_{\tau i+j} \right) \mid \mathcal{F}_{\tau i+j} \right] \right)$$
$$+ z_{\tau i+j} \eta \left( \ell_{\tau(i+1)+j} \left( w_{\text{ref}} \right) - \ell_{\tau(i+1)+j} \left( w_{\tau i+j} \right) \right) - 8 \left( z_{\tau i+j} \eta \right)^2 C_4^2 \mathbf{D}_\psi \left( w_{\text{ref}}; w_{\tau i+j} \right),$$
$$\forall 0 \leq i \leq \frac{T-1-j}{\tau}$$
$$S_k^j = \sum_{i=0}^{k} Z_i^j, \forall 0 \leq k \leq \frac{T-1-j}{\tau}$$

where

$$z_t = \frac{1}{8\eta^2 C_4^2 \left( T + 1 + t \right)} \qquad \forall -1 \leq t \leq T-1$$

We bound

$$\left| \mathbb{E} \left[ \ell_{\tau(i+1)+j} \left( w_{\text{ref}} \right) \mid \mathcal{F}_{\tau i+j} \right] - \mathbb{E} \left[ \ell_{\tau(i+1)+j} \left( w_{\tau i+j} \right) \mid \mathcal{F}_{\tau i+j} \right] \right.$$
$$\left. + \ell_{\tau(i+1)+j} \left( w_{\text{ref}} \right) - \ell_{\tau(i+1)+j} \left( w_{\tau i+j} \right) \right|$$
$$\leq 2C_4 \left\| w_{\text{ref}} - w_{\tau i+j} \right\|$$

By Lemma 15

$$\mathbb{E} \left[ \exp \left( Z_i^j \right) \mid \mathcal{F}_{\tau i+j} \right]$$
$$= \exp \left( -8 \left( z_{\tau i+j} \eta \right)^2 C_4^2 \mathbf{D}_\psi \left( w_{\text{ref}}; w_{\tau i+j} \right) \right)$$
$$\times \mathbb{E} \left[ \exp \left( z_{\tau i+j} \eta \left( \mathbb{E} \left[ \ell_{\tau(i+1)+j} \left( w_{\text{ref}} \right) \mid \mathcal{F}_{\tau i+j} \right] - \mathbb{E} \left[ \ell_{\tau(i+1)+j} \left( w_{\tau i+j} \right) \mid \mathcal{F}_{\tau i+j} \right] \right. \right. \right.$$
$$\left. \left. \left. + \ell_{\tau(i+1)+j} \left( w_{\text{ref}} \right) - \ell_{\tau(i+1)+j} \left( w_{\tau i+j} \right) \right) \right) \mid \mathcal{F}_{\tau i+j} \right]$$
$$\leq \exp \left( -8 \left( z_{\tau i+j} \eta \right)^2 C_4^2 \mathbf{D}_\psi \left( w_{\text{ref}}; w_{\tau i+j} \right) \right) \exp \left( 4 \left( z_{\tau i+j} \eta \right)^2 C_4^2 \left\| w_{\text{ref}} - w_{\tau i+j} \right\|^2 \right) \leq 1$$

Therefore $\mathbb{E} \left[ \exp \left( Z_i^j \right) \mid \mathcal{F}_{\tau i+j} \right] \leq 1$ and hence $\left( \exp \left( S_k^j \right) \right)_{k \geq 0}$ is a supermartingale. By Ville's inequality, we have with probability at least $1 - \delta$, for all $0 \leq k \leq \kappa$

$$\sum_{i=0}^{k} Z_i^j \leq \log \frac{1}{\delta}$$

By union bound over $j = 0, \ldots, \tau - 1$, and with probability at least $1 - \tau\delta$ we have for all $0 \leq k \leq T - \tau - 1$

$$\sum_{t=0}^{k} z_t \eta \left( \mathbb{E} \left[ \ell_{t+\tau} \left( w_{\text{ref}} \right) \mid \mathcal{F}_t \right] - \mathbb{E} \left[ \ell_{t+\tau} \left( w_t \right) \mid \mathcal{F}_t \right] + \ell_{t+\tau} \left( w_{\text{ref}} \right) - \ell_{t+\tau} \left( w_t \right) \right)$$

$$\leq \sum_{t=0}^{k} 8 \left(z_t \eta\right)^2 C_4^2 \mathbf{D}_\psi \left(w_{\text{ref}}; w_t\right) + \tau \log \frac{1}{\delta}$$

We will proceed to prove by induction that $\mathbf{D}_\psi \left(w_{\text{ref}}; w_t\right) \leq \frac{1}{2} R^2$

For the base case $t = 0$, this holds trivially. Suppose that this is true for all $0 \leq t \leq k$, we now show for $t = k + 1$.

If $k \leq \tau - 1$,

$$\sum_{t=0}^{k} \eta \left(\mathcal{R}(w_t) - \mathcal{R}^*\right) + \mathbf{D}_\psi \left(w_{\text{ref}}; w_{k+1}\right)$$

$$\leq D_0 + \sum_{t=0}^{k} \eta \left(\mathcal{R}(w_{\text{ref}}) - \mathcal{R}^*\right)$$

$$+ \sum_{t=0}^{k} \eta \left(\mathcal{R}(w_t) - \mathcal{R}(w_{\text{ref}}) + \ell_{t+1}\left(w_{\text{ref}}\right) - \ell_{t+1}\left(w_t\right)\right) + \frac{\eta^2}{2} \sum_{t=0}^{k} \|g_{t+1}\|_*^2$$

We have

$$\sum_{t=0}^{k} \eta \left|\mathcal{R}(w_t) - \mathcal{R}(w_{\text{ref}}) + \ell_{t+1}\left(w_{\text{ref}}\right) - \ell_{t+1}\left(w_t\right)\right|$$

$$\leq \sum_{t=0}^{k} 2\eta C_4 \|w_{\text{ref}} - w_t\| \leq 2\eta C_4 R(k+1) \leq 2\eta C_4 R\tau$$

and

$$\frac{\eta^2}{2} \sum_{t=0}^{k} \|g_{t+1}\|_*^2 \leq \frac{\eta^2}{2} \sum_{t=0}^{k} \left(C_1 + C_2 \left(1 + \|w_t\|\right)\right)^2$$

$$\leq \frac{\eta^2}{2} \sum_{t=0}^{k} \left(C_4 + C_2 \|w_t - w_{\text{ref}}\|\right)^2$$

$$\leq \eta^2 C_4^2 (k+1) + \eta^2 C_2^2 R^2 (k+1)$$

$$\leq \tau \eta^2 \left(C_4^2 + C_2^2 R^2\right)$$

Therefore

$$\mathbf{D}_\psi \left(w_{\text{ref}}; w_{k+1}\right) \leq D_0 + \eta D_0 \sqrt{T} + 2\eta C_4 R\tau + \tau \eta^2 \left(C_4^2 + C_2^2 R^2\right) \leq \frac{R^2}{2}.$$

If $k \geq \tau$,

$$\sum_{t=0}^{k} z_t \eta \left(\mathcal{R}(w_t) - \mathcal{R}^*\right) + z_k \mathbf{D}_\psi \left(w_{\text{ref}}; w_{k+1}\right) - z_{-1} \mathbf{D}_\psi \left(w_{\text{ref}}; w_0\right)$$

$$\leq \sum_{t=0}^{k} z_t \eta \left(\mathcal{R}(w_{\text{ref}}) - \mathcal{R}^*\right) + \sum_{t=0}^{k} z_t \eta \left(\mathcal{R}(w_t) - \mathcal{R}(w_{\text{ref}})\right)$$

$$+ \sum_{t=0}^{k} z_t \eta \left(\ell_{t+1}\left(w_{\text{ref}}\right) - \ell_{t+1}\left(w_t\right)\right) + \sum_{t=0}^{k} \frac{z_t \eta^2}{2} \|g_{t+1}\|_*^2 + \sum_{t=0}^{k} \left(z_t - z_{t-1}\right) \mathbf{D}_\psi \left(w_{\text{ref}}; w_t\right)$$

$$\leq \sum_{t=0}^{k} z_t \eta \left(\mathcal{R}(w_{\text{ref}}) - \mathcal{R}^*\right) + \sum_{t=k-\tau+1}^{k} z_t \eta \left(\mathcal{R}(w_t) - \mathcal{R}(w_{\text{ref}})\right)$$

$$+ \sum_{t=0}^{k-\tau} z_t \eta \left(\mathcal{R}(w_t) - \mathcal{R}(w_{\text{ref}}) - \mathbb{E}\left[\ell_{t+\tau}\left(w_{\text{ref}}\right) \mid \mathcal{F}_t\right] + \mathbb{E}\left[\ell_{t+\tau}\left(w_t\right) \mid \mathcal{F}_t\right]\right)$$

$$+ \sum_{t=0}^{k-\tau} z_t \eta \left( \mathbb{E}\left[ \ell_{t+\tau}\left(w_{\mathrm{ref}}\right) \mid \mathcal{F}_t \right] - \mathbb{E}\left[ \ell_{t+\tau}\left(w_t\right) \mid \mathcal{F}_t \right] + \ell_{t+\tau}\left(w_{\mathrm{ref}}\right) - \ell_{t+\tau}\left(w_t\right) \right)$$

$$+ \sum_{t=0}^{k-\tau} z_t \eta \left( \ell_{t+\tau}\left(w_t\right) - \ell_{t+\tau}\left(w_{\mathrm{ref}}\right) \right) + \sum_{t=0}^{k} z_t \eta \left( \ell_{t+1}\left(w_{\mathrm{ref}}\right) - \ell_{t+1}\left(w_t\right) \right)$$

$$+ \sum_{t=0}^{k-\tau} \left( z_t - z_{t-1} \right) \mathbf{D}_\psi \left( w_{\mathrm{ref}}; w_t \right) + \sum_{t=0}^{k} \frac{z_t \eta^2}{2} \left\| g_{t+1} \right\|_*^2$$

$$\leq \sum_{t=0}^{k} z_t \eta \left( \mathcal{R}(w_{\mathrm{ref}}) - \mathcal{R}^* \right) + \sum_{t=k-\tau+1}^{k} z_t \eta \left( \mathcal{R}(w_t) - \mathcal{R}(w_{\mathrm{ref}}) \right)$$

$$+ \sum_{t=0}^{k-\tau} z_t \eta \left( \mathcal{R}(w_t) - \mathcal{R}(w_{\mathrm{ref}}) - \mathbb{E}\left[ \ell_{t+\tau}\left(w_{\mathrm{ref}}\right) \mid \mathcal{F}_t \right] + \mathbb{E}\left[ \ell_{t+\tau}\left(w_t\right) \mid \mathcal{F}_t \right] \right)$$

$$+ \tau \log \frac{1}{\delta} + \sum_{t=0}^{k-\tau} 8 \left( z_t \eta \right)^2 C_4^2 \mathbf{D}_\psi \left( w_{\mathrm{ref}}; w_t \right) + \sum_{t=0}^{k-\tau} \left( z_t - z_{t-1} \right) \mathbf{D}_\psi \left( w_{\mathrm{ref}}; w_t \right)$$

$$+ \sum_{t=0}^{k} \frac{z_t \eta^2}{2} \left\| g_{t+1} \right\|_*^2 + \sum_{t=0}^{k-\tau} z_t \eta \left( \ell_{t+\tau}\left(w_t\right) - \ell_{t+\tau}\left(w_{t+\tau-1}\right) \right)$$

$$+ \sum_{t=\tau-1}^{k-1} z_{t-\tau+1} \eta \left( \ell_{t+1}\left(w_t\right) - \ell_{t+1}\left(w_{\mathrm{ref}}\right) \right) + \sum_{t=0}^{k} z_t \eta \left( \ell_{t+1}\left(w_{\mathrm{ref}}\right) - \ell_{t+1}\left(w_t\right) \right)$$

$$\leq \sum_{t=0}^{k} z_t \eta \left( \mathcal{R}(w_{\mathrm{ref}}) - \mathcal{R}^* \right) + \sum_{t=k-\tau+1}^{k} z_t \eta \left( \mathcal{R}(w_t) - \mathcal{R}(w_{\mathrm{ref}}) \right)$$

$$+ \sum_{t=0}^{k-\tau} z_t \eta \left( \mathcal{R}(w_t) - \mathcal{R}(w_{\mathrm{ref}}) - \mathbb{E}\left[ \ell_{t+\tau}\left(w_{\mathrm{ref}}\right) \mid \mathcal{F}_t \right] + \mathbb{E}\left[ \ell_{t+\tau}\left(w_t\right) \mid \mathcal{F}_t \right] \right) + \tau \log \frac{1}{\delta}$$

$$+ \sum_{t=0}^{k} \frac{z_t \eta^2}{2} \left\| g_{t+1} \right\|_*^2 + \sum_{t=0}^{k-\tau} z_t \eta \left( \ell_{t+\tau}\left(w_t\right) - \ell_{t+\tau}\left(w_{t+\tau-1}\right) \right)$$

$$+ \sum_{t=\tau-1}^{k-1} \eta \left( z_{t-\tau+1} - z_t \right) \left( \ell_{t+1}\left(w_t\right) - \ell_{t+1}\left(w_{\mathrm{ref}}\right) \right)$$

$$+ \sum_{t=0}^{\tau-2} z_t \eta \left( \ell_{t+1}\left(w_{\mathrm{ref}}\right) - \ell_{t+1}\left(w_t\right) \right) + z_k \eta \left( \ell_{k+1}\left(w_{\mathrm{ref}}\right) - \ell_{k+1}\left(w_k\right) \right)$$

where in the last inequality we use $z_t = \frac{1}{8\eta^2 C_4^2 (T+1+t)}$ to see that $z_t + 8 \left( z_t \eta \right)^2 C_4^2 \leq z_{t-1}$. Notice that, $\frac{z_t}{z_k} \leq 2$, and $\left( \mathcal{R}(w_{\mathrm{ref}}) - \mathcal{R}^* \right) \leq \frac{D_0}{\sqrt{T}}$ we have

$$\sum_{t=0}^{k} \eta \left( \mathcal{R}(w_t) - \mathcal{R}^* \right) + \mathbf{D}_\psi \left( w_{\mathrm{ref}}; w_{k+1} \right)$$

$$\leq 2D_0 + 2\eta D_0 \sqrt{T} + 16\eta^2 C_4^2 T \tau \log \frac{1}{\delta} + 2\eta \underbrace{\sum_{t=k-\tau+1}^{k} \left| \mathcal{R}(w_t) - \mathcal{R}(w_{\mathrm{ref}}) \right|}_{A}$$

$$+ 2\eta \underbrace{\sum_{t=0}^{k-\tau} \left| \mathcal{R}(w_t) - \mathcal{R}(w_{\mathrm{ref}}) - \mathbb{E}\left[ \ell_{t+\tau}\left(w_{\mathrm{ref}}\right) \mid \mathcal{F}_t \right] + \mathbb{E}\left[ \ell_{t+\tau}\left(w_t\right) \mid \mathcal{F}_t \right] \right|}_{B}$$

$$+ \eta^2 \underbrace{\sum_{t=0}^{k} \|g_{t+1}\|_*^2}_{C} + 2\eta \underbrace{\sum_{t=0}^{k-\tau} |\ell_{t+\tau}(w_t) - \ell_{t+\tau}(w_{t+\tau-1})|}_{D}$$

$$+ \underbrace{\frac{2(\tau-1)\eta}{T} \sum_{t=\tau-1}^{k-1} |\ell_{t+1}(w_t) - \ell_{t+1}(w_{\text{ref}})| + 2\eta \sum_{t=0}^{\tau-2} |\ell_{t+1}(w_{\text{ref}}) - \ell_{t+1}(w_t)| + 2\eta |\ell_{k+1}(w_{\text{ref}}) - \ell_{k+1}(w_k)|}_{E}$$

Now we bound each term. For $A$

$$A = 2\eta \sum_{t=k-\tau+1}^{k} |\mathcal{R}(w_t) - \mathcal{R}(w_{\text{ref}})| \leq 2\eta \sum_{t=k-\tau+1}^{k} C_4 \|w_{\text{ref}} - w_t\| \leq 2\eta\tau C_4 R$$

For $B$, by Assumption 3, $\sup_{t \in \mathbb{Z}_{\geq 0}} \sup_{\mathcal{F}_t} \mathrm{TV}\left(P_t^{t+\tau}, \pi\right) \leq \epsilon$,

$$2\eta |\mathcal{R}(w_t) - \mathcal{R}(w_{\text{ref}}) - \mathbb{E}\left[\ell_{t+\tau}(w_{\text{ref}}) \mid \mathcal{F}_t\right] + \mathbb{E}\left[\ell_{t+\tau}(w_t) \mid \mathcal{F}_t\right]| \leq 2\eta C_4 R\epsilon$$

Thus

$$B = 2\eta \sum_{t=0}^{k-\tau} |\mathcal{R}(w_t) - \mathcal{R}(w_{\text{ref}}) - \mathbb{E}\left[\ell_{t+\tau}(w_{\text{ref}}) \mid \mathcal{F}_t\right] + \mathbb{E}\left[\ell_{t+\tau}(w_t) \mid \mathcal{F}_t\right]| \leq 2\eta C_4 R\epsilon T$$

For $C$, similarly to before

$$C = \eta^2 \sum_{t=0}^{k} \|g_{t+1}\|_*^2 \leq 2T\eta^2 \left(C_4^2 + C_2^2 R^2\right)$$

For $D$, we have

$$|\ell_{t+\tau}(w_t) - \ell_{t+\tau}(w_{t+\tau-1})|$$

$$\leq \sum_{i=t+1}^{t+\tau-1} |\ell_{t+\tau}(w_i) - \ell_{t+\tau}(w_{i-1})|$$

$$\leq \sum_{i=t+1}^{t+\tau-1} \|w_i - w_{i-1}\| \left(C_1 + C_2\left(1 + \|w_i\|\right)\right)$$

$$\leq \sum_{i=t+1}^{t+\tau-1} \eta \|\nabla \ell_i(w_{i-1})\| \left(C_4 + C_2 \|w_i - w_{\text{ref}}\|\right)$$

$$\leq \eta \left(C_4 + C_2 R\right) \sum_{i=t+1}^{t+\tau-1} \left(C_4 + C_2 \|w_{i-1} - w_{\text{ref}}\|\right)$$

$$\leq \eta \left(C_4 + C_2 R\right)^2 \tau \leq \eta\tau \left(2C_4^2 + 2C_2^2 R^2\right)$$

We obtain

$$D = 2\eta \sum_{t=0}^{k-\tau} |\ell_{t+\tau}(w_t) - \ell_{t+\tau}(w_{t+\tau-1})| \leq 2\eta^2 \tau T \left(2C_4^2 + 2C_2^2 R^2\right)$$

For $E$, since

$$|\ell_{t+1}(w_t) - \ell_{t+1}(w_{\text{ref}})| \leq C_4 R$$

Hence

$$E = \frac{2(\tau-1)\eta}{T} \sum_{t=\tau-1}^{k-1} |\ell_{t+1}(w_t) - \ell_{t+1}(w_{\text{ref}})|$$

$$+ 2\eta \sum_{t=0}^{\tau-2} |\ell_{t+1}(w_{\text{ref}}) - \ell_{t+1}(w_t)| + 2\eta |\ell_{k+1}(w_{\text{ref}}) - \ell_{k+1}(w_k)|$$

$$\leq 2\eta C_4 R \left( \frac{(\tau - 1)(k - \tau + 1)}{T} + \tau \right) \leq 4\eta\tau C_4 R$$

Sum up we have

$$\sum_{t=0}^{k} \eta \left( \mathcal{R}(w_t) - \mathcal{R}^* \right) + \mathbf{D}_\psi \left( w_{\text{ref}}; w_{k+1} \right)$$

$$\leq 2D_0 + 2\eta D_0 \sqrt{T} + 16\eta^2 C_4^2 T\tau \log \frac{1}{\delta}$$
$$+ 2\eta\tau C_4 R + 2\eta C_4 R\epsilon T + 2T\eta^2 \left( C_4^2 + C_2^2 R^2 \right)$$
$$+ 2\eta^2 \tau T \left( 2C_4^2 + 2C_2^2 R^2 \right) + 4\eta\tau C_4 R$$

$$\leq \frac{R^2}{2}$$

as needed. Finally we have

$$\frac{1}{k+1} \sum_{t=0}^{k} \left( \mathcal{R}(w_t) - \mathcal{R}^* \right) + \frac{\mathbf{D}_\psi \left( w_{\text{ref}}; w_{k+1} \right)}{\eta (k+1)} \leq \frac{R^2}{2\eta (k+1)}.$$

$\square$

