\left\| \nabla \ell_i \left( w_{i-1} \right) \right\| \left( C_4 + C_2 \left\| w_i - w_{\text{ref}} \right\| \right)$$

$$\leq \eta \left( C_4 + C_2 R \right) \sum_{i=t+1}^{t+\tau-1} \left( C_4 + C_2 \left\| w_{i-1} - w_{\text{ref}} \right\| \right)$$

$$\leq \eta \left( C_4 + C_2 R \right)^2 \tau \leq \eta \tau \left( 2C_4^2 + 2C_2^2 R^2 \right)$$

We obtain

$$D = 2\eta \sum_{t=0}^{k-\tau} \left| \ell_{t+\tau} \left( w_t \right) - \ell_{t+\tau} \left( w_{t+\tau-1} \right) \right| \leq 2\eta^2 \tau T \left( 2C_4^2 + 2C_2^2 R^2 \right)$$

For $E$, since

$$\left| \ell_{t+1} \left( w_t \right) - \ell_{t+1} \left( w_{\text{ref}} \right) \right| \leq C_4 R$$

Hence

$$E = \frac{2(\tau-1)\eta}{T} \sum_{t=\tau-1}^{k-1} \left| \ell_{t+1} \left( w_t \right) - \ell_{t+1} \left( w_{\text{ref}} \right) \right|$$

$$+ 2\eta \sum_{t=0}^{\tau-2} \left| \ell_{t+1} \left( w_{\text{ref}} \right) - \ell_{t+1} \left( w_t \right) \right| + 2\eta \left| \ell_{k+1} \left( w_{\text{ref}} \right) - \ell_{k+1} \left( w_k \right) \right|$$

$$\leq 2\eta C_4 R \left( \frac{(\tau-1)(k-\tau+1)}{T} + \tau \right) \leq 4\eta \tau C_4 R$$

Sum up we have

$$\sum_{t=0}^{k} \eta \left( \mathcal{R}(w_t) - \mathcal{R}^* \right) + \mathbf{D}_\psi \left( w_{\text{ref}}; w_{k+1} \right)$$

$$\leq 2D_0 + 2\eta D_0 \sqrt{T} + 16\eta^2 C_4^2 T \tau \log \frac{1}{\delta}$$

$$+ 2\eta \tau C_4 R + 2\eta C_4 R \epsilon T + 2T\eta^2 \left( C_4^2 + C_2^2 R^2 \right)$$

$$+ 2\eta^2 \tau T \left( 2C_4^2 + 2C_2^2 R^2 \right) + 4\eta \tau C_4 R$$

$$\leq \frac{R^2}{2}$$

as needed. Finally we have

$$\frac{1}{k+1} \sum_{t=0}^{k} \left( \mathcal{R}(w_t) - \mathcal{R}^* \right) + \frac{\mathbf{D}_\psi \left( w_{\text{ref}}; w_{k+1} \right)}{\eta \left( k+1 \right)} \leq \frac{R^2}{2\eta \left( k+1 \right)}.$$

$\square$