# OpenReview forum: "On the Generalization Error of Stochastic Mirror Descent for Quadratically-Bounded Losses: an Improved Analysis"
_NeurIPS.cc/2023/Conference — NeurIPS 2023 poster_

### Official Review · Reviewer_sY4Q · 2023-06-29

**Soundness:** 3 good
**Presentation:** 3 good
**Contribution:** 3 good
**Rating:** 6
**Confidence:** 3

**Summary:**

This paper analyzes the generalization bound for stochastic mirror descent applied to linear models and a general class of quadratically-bounded losses. In addition to the more common IID sampled data, this paper also considers where the data is sampled from a Markov chain. This paper follows up from the work of [Telgarsky 2022] and improved the bounds, particularly in the case of non-realizable setting. The improvements in this paper come from a more clever application of concentration inequalities.

**Strengths:**

Since this paper is mainly a follow up to [Telgarsky, 2022], I would refer that paper as [T22] for simplicity.

One major strength of this paper compare to [T22] is that the content are better organized and the notations are also slightly better. I found this paper to be generally easy to follow.

Also, the theoretical analysis seem to be in good shape. While I did not have the time to read through the proofs line-by-line, I have reasonable faith that there are not major technical issues buried in the proof.

**Weaknesses:**

While the overall quality of this paper is good, I feel that the contribution over [T22] is not sufficient to warrant a new paper. In particular, compare to [T22]: Theorems 3 and 13 in this paper have the same guarantee as Theorems 5 and 8 in [T22], respectively. While Theorem 7 of this paper is an improvement over Theorem 10 of [T22], a log factor is not particularly interesting. Also, the problem setting of this paper is almost entirely the same as [T22] and I would love to see more novelty on this front.

**Questions:**

I would like to hear from the author on how this paper presents a significant delta over [T22] and in what ways their improved bounds can be useful. Given what I currently gathered from the paper, I am inclined to say that the contribution is insufficient, but I would love to be proven wrong.

**Limitations:**

N/A. This paper is purely theoretical and I do not believe there would be any negative society impact.

---

> ### Author Rebuttal · Authors · 2023-08-09
>
> We thank the reviewer for the valuable feedback.
>
> **Regarding the comparison between our results and [T22]**: Both theorems 3 and 13 in our paper improve the guarantee of theorems 5 and 8 in [T22] by a $\log T$ factor. This is also the case of Theorem 7 compared with Theorem 10 part 1 for sub-gaussian data in [T22]. Our Theorem 10 significantly improves Theorem 10 (part 2 for polynomial tailed data) in [T22], i.e, by a $poly\ T$ factor. As in our response to reviewer TJAZ, the significance of this improvement lies in two aspects:
>
> 1. in our opinion, closing the gap between the upper bounds and lower bounds is an important question in the literature of optimization and generalization, even to $\log$ factors. Although a $\log$ factor improvement may seem small, there is a line of works studying the $\log$ factors that reveal fascinating phenomena of SGD. For example, [3] shows that the lower bound for the convergence of last iterate of SGD differs from that of the suffix averaging iterate by a $\log$ factor. This theoretical improvement strengthens our understanding of the behavior of SMD. In the non-realizable case with polynomial tailed data, where the improvement by our approach is significant, we also have a much better understanding of the behavior of the algorithm.
>
> 2. we offer new approaches that look into to how obtain a tighter high probability bound, instead of just applying concentration inequalities as a blackbox (which is suboptimal). As noted by Reviewer H14F, our work “depart[s] significantly from the prior work of Telgarsky (2022), and introduce[s] a novel approach for establishing high probability generalization guarantees”. We think our new approaches can be applicable in broader scenarios.
>
> We would also like to point out an issue in the analysis of [T22], which is unclear to us during our study of the paper. The application of Azuma's inequality (mentioned in the last paragraph of page 12 - page 13 in [T22]) in proving Theorem 10 [T22] is not correct (or at least not immediate to us) as the variables' ranges do not satisfy the measurability condition of the inequality. Our analysis fixed this issue in this paper, having to rely on a different approach, while also achieving significant improvements.
>
> **Regarding the problem setting**: As detailed in [T22], the setting studied in this paper captures and significantly extends many prior works in a unified way. Due to its generality, it remains interesting to provide a tight bound for the generalization error in this broad class. There is also large body of work on obtaining generalization bounds via Rademacher complexity. For example, the recent work [1] considers the setting of separable data and smooth losses with bounded tails. We think it is a very challenging and interesting direction for future works to unify these two settings.
>
> Another broad direction for future works is to show generalization bounds for a more general class of algorithms. For example, algorithmic stability was used by [2] to show high probability bounds for Lipschitz losses for several variants of SGD in a unified way. Going beyond SMD, it would be interesting to combine our techniques with [2] to obtain tight high probability generalization bounds for both a more general class of algorithms and all quadratically bounded losses.
>
> Reference:
>
> [1] Schliserman, Matan, and Tomer Koren. "Tight Risk Bounds for Gradient Descent on Separable Data." arXiv:2303.01135 (2023).
>
> [2] Bassily, Raef, et al. "Stability of stochastic gradient descent on nonsmooth convex losses." Advances in Neural Information Processing Systems 33 (2020).

---

> > ### Comment · Reviewer_sY4Q · 2023-08-10
> >
> > Thank you for your responses. I am still not convinced that a $\log T$ factor improvement is significant enough to warrant a new paper. But I am interested in hearing more about some of the technical details you offered:
> >
> > 1. What is reference [3]? I want to check out this work but it is not listed in your response.
> >
> > 2. Can you elaborate how the step in [T22] that you think is incorrect? If your claim is indeed correct, then my valuation of this paper would raise significantly.

---

> > > ### Author Response · Authors · 2023-08-11
> > >
> > > Thank you for your additional comment.
> > >
> > > We are sorry for missing the reference [3] in the rebuttal. It is as follows:
> > >
> > > [3] Harvey, Nicholas JA, et al. "Tight analyses for non-smooth stochastic gradient descent." Conference on Learning Theory. PMLR, 2019.
> > >
> > > **Regarding the issue in [T22]**: This issue arises in the proof of Theorem 10 in [T22] (appendix B.4), which includes both cases of sub-Gaussian tails and polynomial tails in the non-realizable setting with IID data. First let us recall the Azuma-type inequality (van Handel, 2016, Problem 3.11) that is used in [T22]: Let $\{F_{t}\}$ be a filtration and $A_{t},B_{t},\Delta_{t}$ satisfy
> > >
> > > 1. $\Delta_{t}$ is $F_{t}$ measurable and $E\left[\Delta_{t}|F_{t-1}\right]=0$
> > >
> > > 2. $A_{t},B_{t}$ are $F_{t-1}$ measurable and $A_{t}\le\Delta_{t}\le B_{t}$ almost surely
> > >
> > > then if $\sum_{t=1}^{n}(B_{t}-A_{t})^{2}\le c^{2}$ with probability at least $1-\delta$, we have $\sum_{t=1}^{n}\Delta_{t}	\le\sqrt{\frac{c^{2}\log\frac{1}{\delta}}{2}}$.
> > >
> > > In [T22], the authors want to bound for a solution $v_{t}$ projected on a ball of radius R the following
> > >
> > > $$\sum_{t=1}^{n}\ell_{t+1}(w_{ref})-\ell_{t+1}(v_{t})-R(w_{ref})+R(v_{t})$$
> > >
> > > What we know is that $E[\ell_{t+1}(w_{ref})-\ell_{t+1}(v_{t})-R(w_{ref})+R(v_{t})\mid F_{t}]=0$.
> > >
> > > and from Lemma 1 $|\ell_{t+1}(w_{ref})-\ell_{t+1}(v_{t})| \le CZ_{t+1}^{2}$ where C is a constant and $Z_{t+1}=\max\\{\\|x\_{t+1}\\|\_{*},y\_{t+1}\\}$. The authors next use the following bound for the second condition:
> > >
> > > $$-2CZ_{t+1}^{2}\le \ell_{t+1}(w_{ref})-\ell_{t+1}(v_{t})-R(w_{ref})+R(v_{t})\le2CZ_{t+1}^{2}$$
> > >
> > > They further bound $\sum Z_{t}^{4}$ (in probability $1-\delta$), and then apply the above Azuma's inequality to obtain the final bound.
> > >
> > > However, the above bound does not satisfy the second condition in Azuma's inequality, because $Z_{t+1}\notin F_{t}$ i.e. $Z_{t+1}$ is not $F_{t}$ measurable. This is for a fundamental reason: $\\|x\_{t+1}\\|\_{*},y\_{t+1}$ are correlated with $\ell_{t+1}$.
> > >
> > > The proofs of our Theorem 7 for sub-Gaussian tails and Theorem 10 for polynomial tails overcome this using different techniques. In theorem 7, we directly analyze the moment generating function of the martingale sequence. In Theorem 10, we need a more elaborate truncation technique to avoid the correlation between the random variable (the $\Delta_{t}$ above) and the bounds (the $A_{t},B_{t}$ above). A sketch of our proof for theorem 10 can be found at lines 253-263 in the paper.
> > >
> > > Reference:
> > >
> > > Ramon van Handel. Apc 550 lecture notes: Probability in high dimensions, Dec 2016.

---

> > > > ### Comment · Reviewer_sY4Q · 2023-08-12
> > > >
> > > > I don't quit get your point. $x_{t+1}$ and $y_{t+1}$ are sampled from a Markov chain so their distributions are determined from $x_t$ and $y_t$, why would they not be $F_t$-measurable?
> > > >
> > > > I applaud for your attention to detail because many people (myself included) do not check measurability very carefully. But my current feeling is that you overthought this detail?

---

> > > > > ### Author Response · Authors · 2023-08-12
> > > > >
> > > > > Thank you for your response. The issue we mentioned is for the case of IID data in the non-realizable setting (In the proof of Theorem 10 in [T22]). Since $(x_{t+1},y_{t+1})$ are sampled independently from previous iterations, we have the measurability issue, which is a quite fundamental problem and for which we think there is no easy fix. We hope this clarifies that the reviewer's concern.

---

> > > > > > ### Comment · Reviewer_sY4Q · 2023-08-13
> > > > > >
> > > > > > Ah okay, I was confused by the various settings used in this work. While I can see your point now, I do not have the time to verify your proofs line-by-line. I will raise my score to **6** on the condition that *your revision would highlight the issue in [T22]* so the community can assess this more thoroughly.

---

> > > > > > > ### Author Response · Authors · 2023-08-13
> > > > > > >
> > > > > > > We thank the reviewer for the positive feedback. We would highlight this aforementioned issue in the revision of our paper.

---

### Official Review · Reviewer_p99H · 2023-07-03

**Soundness:** 3 good
**Presentation:** 3 good
**Contribution:** 3 good
**Rating:** 7
**Confidence:** 3

**Summary:**

The authors study Stochastic Mirror Descent, and analysis its generalization error under a quadratic loss assumptions, self boundedness of the loss (a relaxation of smoothness), iid or Markovian data, and realizable or non-realizable cases, which amount to assume that the error of the reference point is of different orders.
The approach taken consists in deriving concentration bounds taylored to the problem.
The results improve previous ones by some log(T) (resp. polylog(T) ) factor in the realizatble (resp. non-realizable) setting.

**Strengths:**

**Disclaimer.** I acknowledge not being familiar with the generalization error literature cited in the paper (excpet the Hardt et al. paper on algorithmic stability), although I am quite familiar with the SGD literature and especially Mirror descent and its stochastic counterparts.

Stochastic Mirror Descent is a widely used and studied algorithm in many domains of optimization. Since existing generalization error analyses (the SOTA rates being Telgarsky [30]), as cited in the paper) appear to be suboptimal in light of the rates obtained in the present submission, I think this is a contribution worth acceptance at Neurips. The whole paper being organized around the generalization bounds for SMD under the various data assumptions (realizable or not cases, Markovian or iid), it is quite technical, but the authors make an effort to put as much detail as possible of the analysis in the main part of the paper

**Weaknesses:**

I think that the results should be put in light of optimization error analyses of SMD, the current best rates being in the following paper [A]:

**[A]** *Fast Stochastic Bregman Gradient Methods: Sharp Analysis and Variance Reduction*, Dragomir et al., Icml 2021.

The following work also studies SMD convergence rates.

**[B]** *Stochastic Mirror Descent: Convergence Analysis and Adaptive Variants via the Mirror Stochastic Polyak Stepsize*, Loizou et al. 2022.


In particular, the rates in [A] being for SMD with iid data and fresh samples at each iteration (and thus the loss being optimized in the generalization error), can these rates be considered, in the context of the present submission, as in-expectation gen error ?

Also, in many applications, MD or SMD is used when (S)GD cannot converge (or is too slow) due to non-smoothness and degenerate frontiers (e.g, minimization of KL distances). It seems to me that the assumptions on the loss in the present submission are too restrictive to allow this. This, in addition to some strong assumptions on the data lead to somewhat quite restrictive assumptions; this is however fine as it appears that previous works also make similar assumptions. To what extent the gap in the theory between SMD convergence rates in expectation (as in [A]) and the high probability generalization error of this submission seems hard to fill ?

**Questions:**

See above.

I suggest acceptance for this paper for the reasons noted above. I suggest adding references such as [A,B] in order to add discussions for readers who are more familiar with such works.

**Limitations:**

No such limitations.

---

> ### Author Rebuttal · Authors · 2023-08-09
>
> We thank the reviewer for the valuable feedback and for pointing out the references that we missed in our submission. We will include them in the revision.
>
> **Regarding [A]**: We agree that this paper can be seen in part as giving an in-expectation generalization error bound for a setting that corresponds to the non-realizable setting with  iid data. However, one important distinction between these two settings is
>
> - in [A] and other optimization papers, we commonly impose assumptions on the stochastic gradients (such as bounded variance, sub-gaussian noise, etc)
>
> - in our paper and other generalization papers, we make assumptions on the data instead of the stochastic gradients.
>
> **Regarding the assumptions**: Convergence analysis for SMD even in expectation does not immediately translate to generalization error bounds. One reason for that is because we do not have {\em a priori} bounds on the magnitude of the gradients. A challenging question is how we can translate the assumptions on the data to guarantees on the gradients. It turns out, we can make such guarantees only for certain loss functions, such as logistic loss or exponential tailed losses or the broader class of quadratically bounded losses introduced by Telgarsky [30]. For this class of losses, Lemma 1 can be viewed as an attempt to bound the gradients. Another challenge that arises from Lemma 1 is that these bounds depend on the magnitude of the data and therefore convergence analysis for SMD still does not apply. Our paper fills in the gap by introducing a novel analysis.
>
> Additionally, our paper also shows that in the realizable case, for potentially non-smooth losses, the generalization bound can be better than what is shown by the convergence guarantee ($O(1/T)$ vs $O(1/\sqrt{T})$). We also include bounds for the non-iid case (markovian data).

---

> > ### Comment · Reviewer_p99H · 2023-08-17
> >
> > I thank the authors for their answers and keep my opinion of the paper and my score unchanged.

---

### Official Review · Reviewer_TJAZ · 2023-07-05

**Soundness:** 4 excellent
**Presentation:** 3 good
**Contribution:** 3 good
**Rating:** 6
**Confidence:** 3

**Summary:**

The authors propose a novel approach to analyze the generalization error of Stochastic Mirror Descent (SMD) for quadratically bounded losses. Compared to previous methods, their approach demonstrates improved accuracy by a logarithmic factor when the data is realizable. Additionally, in scenarios where the data is not realizable and has a polynomial tail, their method achieves enhanced accuracy by a polynomial factor of T.

**Strengths:**

1. The paper provides a clear and well-explained improvement over previous work.
2. The approach presented in the paper, which involves analyzing the moment generating function of a martingale difference sequence with well-chosen coefficients is interesting as it avoids the need for the coupling technique used in previous works and might be used also in future works.

**Weaknesses:**

1.A potential weakness of the paper is that the improvement over previous work in the realizable case is relatively modest, as it is limited to logarithmic factors in the bound.
2. Although the paper suggests improvements in the bound, these may not result in entirely new conclusions or fundamentally alter our understanding of the generalization behavior of Stochastic Mirror Descent (SMD) with quadratically bounded losses. The incremental nature of the improvement might not have a significant impact on the practical performance or broader understanding of the algorithm.

**Questions:**

See weaknesses.

**Limitations:**

yes

---

> ### Author Rebuttal · Authors · 2023-08-09
>
> We thank the reviewer for the feedback. We answer the concerns of the reviewer below.
>
> 1. The main contribution in the realizable case is not only an improvement in the log factors, but also a new approach to analyze generalization bounds in high probability of stochastic gradient methods. As noted by Reviewer H14F, our work “depart[s] significantly from the prior work of Telgarsky (2022), and introduce[s] a novel approach for establishing high probability generalization guarantees”. We think our new approaches can be applicable in broader scenarios.
>
> 2. To our knowledge, closing the gap between the upper bounds and lower bounds is an important question in the literature of optimization and generalization. even to $\log$ factors. Although a $\log$ factor improvement may seem small, there is a line of works studying the log factors that reveal fascinating phenomena of SGD. For example, [1] shows that the lower bound for the convergence of last iterate of SGD differs from that of the suffix averaging iterate by a $\log$ factor. This theoretical question does not fundamentally change the performance of the algorithm we use in practice, but strengthens our understanding. In the non-realizable case with polynomial tailed data, where the improvement by our approach is significant, we also have a much better understanding of the behavior of the algorithm.
>
> Reference
>
> [1] Harvey, Nicholas JA, et al. "Tight analyses for non-smooth stochastic gradient descent." Conference on Learning Theory. PMLR, 2019.

---

> > ### Comment · Reviewer_TJAZ · 2023-08-14
> > **Answer to the rebuttal**
> >
> > Thanks to the authors for addressing my concerns and having a detailed discussion about the possible issue in [T22]. Such discussion should definitely be taken into account for the next revisions of the paper.

---

### Official Review · Reviewer_H14F · 2023-07-09

**Soundness:** 3 good
**Presentation:** 3 good
**Contribution:** 3 good
**Rating:** 5
**Confidence:** 3

**Summary:**

This paper revisits the generalization error of stochastic mirror descent for quadratically bounded losses studied in Telgarsky (2022). The authors depart significantly from the prior work of Telgarsky (2022), and introduce a novel approach for establishing high probability generalization guarantees. In contrast to the prior work, this work directly analyzes the moment generating function of a novel supermartingale sequence and leverages the structure of stochastic mirror descent. As a result, this paper obtains improved bounds.

**Strengths:**

This paper revisits the generalization error of stochastic mirror descent for quadratically bounded losses studied in Telgarsky (2022). The authors depart significantly from the prior work of Telgarsky (2022), and introduce a novel approach for establishing high probability generalization guarantees. In contrast to the prior work, this work directly analyzes the moment generating function of a novel supermartingale sequence and leverages the structure of stochastic mirror descent. As a result, this paper obtains improved bounds.

**Weaknesses:**

No

**Questions:**

In the proof of  Lemma 4, line391-line392,  the last inequality uses assumption 2.
However, Assumption assumes that $\ell'(z)^2 \le 2 \ell(z)$ for $z\in \R$ but $w_t$ in the proof is a vector.
Could you fix it?

**Limitations:**

No.

---

> ### Author Rebuttal · Authors · 2023-08-09
>
> We thank the reviewer for the feedback and the endorsement of our novel approaches.
>
> Regarding the question: We believe that line 391-line 392 are correct and consistent with our notations. We recall some notations to clarify the assumptions. We use $\ell_{t}(w)$ as the short form for $\ell(y_{t},w^{T}x_{t})$ and $\ell(y,\widehat{y})$ is either $\widetilde{\ell}(sign(y)\widehat{y})$ or $ \widetilde{\ell}(y-\widehat{y})$. In assumption 2, we use the scalar form $\widetilde{\ell}$ of $\ell$, which says, $\widetilde{\ell}'(z)^{2}\le2\rho\widetilde{\ell}(z)$ for $z\in R$. Regarding line 391-line 392, let us take the first case $\ell(y,\widehat{y})=\widetilde{\ell}(sign(y)\widehat{y})$ as an example (same steps hold for the other case), we have
>
> $||g_{t+1}||\_{*}^2 $
>
> $= || \ell' (y_{t+1},w_{t}^{T}x\_{t+1} )||\_{*}^{2}   $
>
> $\overset{(a)}{=} || x_{t+1} ||\_{*}^2 \widetilde{\ell}' (sign(y_{t+1}) w_{t}^{T} x_{t+1})^2 $
>
> $
> 	\overset{(b)}{\le} 2\rho\widetilde{\ell}(sign(y_{t+1})w_{t}^{T}x_{t+1})$
>
> $=2\rho\ell(y_{t+1},w_{t}^{T}x_{t+1}) $
>
> $=2\rho\ell_{t+1}(w_{t})$
>
> where $\ell'$ denotes a subgradient in $\partial\ell$ wrt the second argument, (a) uses chain rule and (b) uses the assumption $\left\Vert x_{t+1}\right\Vert _{*}\le1$.

---

> > ### Author Response · Authors · 2023-08-19
> > **Did we address all the concerns of the reviewer?**
> >
> > We hope that we have addressed all the concerns of the reviewer. We are happy to answer any additional questions but the author-reviewer discussion period is closing soon. As the reviewer noted, our paper "depart significantly from the prior work" and "introduce a novel approach for establishing high probability generalization guarantees." Given all these positive strengths and no weaknesses, could the reviewer please consider changing the score to align with the strengths of the paper? Thank you!

---

### Decision · Program_Chairs · 2023-09-21

**Decision:**

Accept (poster)

**Comment:**

While the final results of the paper might seem incremental given the prior work (removing poly(T) factors), all the reviewers (and myself) appreciated the introduction of the new proof technique and the fact that the state of the art bounds have improved. Besides the authors have adequately addressed the reviewers' concerns, which resulted in an increase in the overall score. After the discussion with the SAC, we believe that the paper is borderline and we recommend a weak acceptance.